# Chromosome-level genome assembly of hadal snailfish reveals mechanisms of deep-sea adaptation in vertebrates

Wenjie Xu[1†], Chenglong Zhu[1†], Xueli Gao[1†], Baosheng Wu[1†], Han Xu[2†], Mingliang Hu[1†], Honghui Zeng[3†], Xiaoni Gan[3], Chenguang Feng[1], Jiangmin Zheng[1], Jing Bo[2], Li-Sheng He[2], Qiang Qiu[1], Wen Wang[1], Shunping He[1,2,3*], Kun Wang[1*]

[1]School of Ecology and Environment, Northwestern Polytechnical University, Xi'an, China; [2]Institute of Deep-Sea Science and Engineering, Chinese Academy of Sciences, Sanya, China; [3]Key Laboratory of Aquatic Biodiversity and Conservation, Institute of Hydrobiology, Chinese Academy of Sciences, Wuhan, China

*For correspondence:
clad@idsse.ac.cn (SH);
wk8910@gmail.com (KW)

†These authors contributed equally to this work

Competing interest: The authors declare that no competing interests exist.

## Abstract

As the deepest vertebrate in the ocean, the hadal snailfish (*Pseudoliparis swirei*), which lives at a depth of 6,000–8,000 m, is a representative case for studying adaptation to extreme environments. Despite some preliminary studies on this species in recent years, including their loss of pigmentation, visual and skeletal calcification genes, and the role of trimethylamine N-oxide in adaptation to high-hydrostatic pressure, it is still unknown how they evolved and why they are among the few vertebrate species that have successfully adapted to the deep-sea environment. Using genomic data from different trenches, we found that the hadal snailfish may have entered and fully adapted to such extreme environments only in the last few million years. Meanwhile, phylogenetic relationships show that they spread into different trenches in the Pacific Ocean within a million years. Comparative genomic analysis has also revealed that the genes associated with perception, circadian rhythms, and metabolism have been extensively modified in the hadal snailfish to adapt to its unique environment. More importantly, the tandem duplication of a gene encoding ferritin significantly increased their tolerance to reactive oxygen species, which may be one of the important factors in their adaptation to high-hydrostatic pressure.

## eLife assessment

This **important** study advances our understanding of the potential mechanisms of deep-sea adaptation and sheds light on the evolutionary history of hadal snailfish. Through comparative genomic analysis, the authors provide **convincing** evidence and propose hypotheses on the timing of trench colonization, population structure, and adaptations to the hadal snailfish genome in response to their environment.

## Introduction

Since its capture in 2014, the hadal snailfish (*Pseudoliparis swirei*) has captured the attention of biologists and public as the deepest known vertebrate on Earth (*Fujii et al., 2010*; *Gerringer et al., 2017a*; *Gerringer et al., 2021b*). The hadal zone, 6 km below the sea surface where this species lives, and after which the species is named, is characterized by high-hydrostatic pressure (HHP), complete darkness, and barrenness (*Jamieson, 2015*; *Jamieson, 2011*). This special organism that survives and thrives in the hadal zone provides us with an unusual case of adaptation to an extreme environment. After several years of anatomical and genomic research (*Gerringer, 2019*; *Gerringer et al., 2017b*;

*Wang et al., 2019*), it is now known that the hadal snailfish has degraded vision and a lack of melanin in its skin (*Wang et al., 2019*). Previous studies have also revealed that having more unsaturated fatty acids in the cell membrane (*Cossins and MacDonald, 1984*; *Fang et al., 2000*) and high levels of trimethylamine N-oxide (TMAO) in the body (*Yancey et al., 2014*) may have played important roles in resistance to HHP. However, much remains to be discovered about this species; our lack of knowledge can be categorized into three principal areas.

The first concerns the origin of hadal snailfish. They have been observed in several trenches in the northwest Pacific Ocean, including the Mariana (*Wang et al., 2019*), Yap (*Gerringer et al., 2021a*), Kuril–Kamchatka (*Gerringer et al., 2017b*), and Japan (*Gerringer et al., 2021a*) trenches. The question arises: Do they have the ability to migrate across trenches, or do they enter the hadal zone independently? Furthermore, it has been shown that the divergence time between hadal snailfish and Tanaka's snailfish (a closely related species distributed in shallow areas) is about 20 million years ago (Mya) *Wang et al., 2019*; but when did the hadal snailfish enter the hadal zone and how long did they take to complete their adaptation to this ecological niche? The second aspect concerns its morphological and physiological characteristics. For example, in such a dark environment, what do hadal snailfish rely on to sense the world: is it smell, taste, or something else? Do they still have circadian rhythms in the absence of sunlight? Does darkness and HHP have any effect on their behavior? The last area concerns the mechanisms by which they tolerate HHP. If unsaturated fatty acids and TMAO are common in marine fish, why have only a few species such as hadal snailfish been observed to reach such depths? Some studies suggest that certain genetic alterations may confer tolerance to HHP (*Wang et al., 2019*), but if so, how do these alterations help hadal snailfish to adapt to this environment, and which alterations are most critical?

Unfortunately, these questions have not been well resolved because it is difficult for us to make long-term observations of deep-sea organisms in situ. During 2018–2019, we collected multiple samples of hadal snailfish from the Mariana Trench and Tanaka's snailfish from the Yellow Sea. Based on more data and more refined genome, we have been able to trace the genomic signals left by adaptive evolution in an attempt to more fully understand the evolutionary processes and key changes in this special organism.

## Results

### Improved genome assembly for Mariana hadal snailfish

A total of four hadal snailfish (*P. swirei*) and four Tanaka's snailfish (*Liparis tanakae*) individuals were collected for this study (*Supplementary file 1*). Using a combination of Oxford Nanopore Technologies (ONT) long reads, Beijing Genomics Institute (BGI) short reads, and Hi-C sequencing technologies, we generated a chromosome-level genome assembly for hadal snailfish (*Supplementary file 2*). The genome assembly comprised 1,173 contigs (total length = 626.44 Mb, contig N50 = 4.22 Mb), organized into 24 chromosomes with an anchoring rate of 98.24%. The new assembly filled 1.26 Mb of gaps that were present in our previous assembly and have a much higher level of genome continuity and completeness (with complete BUSCOs of 96.0% [Actinopterygii_odb10 database]) than the two previous assemblies (*Figure 1—figure supplements 1 and 2*; *Supplementary file 3 and 4*; *Mu et al., 2021*; *Wang et al., 2019*). Moreover, the genome redundancy caused by mis-assembly is also largely reduced in the new assembly (*Figure 1—figure supplement 3*), which ensures the reliability of the subsequent analysis. Meanwhile, we generated a high-quality chromosomal-level genome assembly for Tanaka's snailfish for a comparative evolutionary study. We noticed that there is no major chromosomal rearrangement between hadal snailfish and Tanaka's snailfish, and chromosome numbers are consistent with the previously reported MTZ-ancestor (the last common ancestor of medaka, *Tetraodon*, and zebrafish) (*Kasahara et al., 2007*), while the stickleback had undergone several independent chromosomal fusion events (*Figure 1—figure supplement 4*).

Based on the new genome assemblies, we re-examined the genetic changes that occurred in the common ancestor of hadal snailfish in combination with the new resequencing and transcriptome data. After a thorough scan and careful inspection, we identified 51 absent genes, 20 unitary pseudogenes, 21 lineage-specific expanded genes, 33 genes with insertions and deletions (with a length ≥3 amino acids) in coding regions, and 33 de novo-originated new genes (*Supplementary file 8-12*). Most of them have not been previously reported and we discuss them in the following sections.

## Cross-trench distribution and high level of genetic diversity

Combining the eight new sequenced individuals (four hadal snailfish and four Tanaka's snailfish) with five previously reported individuals (four hadal snailfish and one Tanaka's snailfish), we have been able to form an initial perspective of the hadal snailfish at the population level. The principal component analysis (PCA), neighbor-joining tree, and genetic clustering analysis show that the eight hadal snailfish individuals can be divided into two populations, the first with seven individuals and the second with one individual (*Figure 1A–C*; *Supplementary file 1*). Interestingly, the first population includes samples from both the Mariana and Yap trenches. Using the mitochondrial data, we found that the divergence time of these individuals from different trenches appears to be only about 44,000 years (*Figure 1—figure supplement 5*). Combined with additional publicly available mitochondrial data, we noticed that the sample from the Kermadec Trench (*Gerringer et al., 2017b*), about 6400 km away from the Mariana Trench, is also clustered with individuals from the first population, and the divergence time was estimated to be 1.0 Mya (*Figure 1D*, *Figure 1—figure supplement 6*). These results suggest that hadal snailfish have successfully spread to multiple trenches in the Pacific Ocean over the course of a million years. And this dispersal may have been caused by population expansion or deep circulation.

Genetic diversity of hadal snailfish is about 3.48 times higher than Tanaka's snailfish. The $F_{ST}$ between the two species is close to 0.91, indicating a large genetic divergence (*Figure 1—figure supplement 7*). That most polymorphisms are unique to each species, and that they share only 6% of their SNPs, is consistent with this observation. In addition, we also estimated the demographic history of hadal snailfish at the population level and observed a significant expansion in the last 60,000 years (*Figure 1—figure supplement 8*).

## Preserved *rh1* gene suggests rapid adaptation to hadal zone

Based on the mitochondrial data, the closest known species related to the hadal snailfish were found to be from the genera *Careproctus*, *Crystallias*, *Rhodichthys,* and *Paraliparis*, which contain many species living at approximately 1,000 m depth (*Figure 1D*; *Supplementary file 13*). The divergence time between hadal snailfish and these species was estimated to be about 9.9 Mya, close to the formation time (8 Mya) of the deepest trough of the Mariana Trench (*Oakley et al., 2009*). We might therefore speculate that the ancestor of hadal snailfish adapted to the deep-sea environment around 1,000 m at about 9.9 Mya, and subsequently gradually adapted to greater depths in the formed or forming trench. The upper and lower limits of the time for hadal snailfish to enter the hadal zone were estimated to be 9.9 Mya (divergence time between hadal snailfish and its closest relatives) and 1.0 Mya (divergence time between different hadal snailfish individuals), respectively. Interestingly, the vision-related genes also confirm a rapid adaptation to hadal zone.

Fish that inhabit different depths of the sea rely on different vision-related genes (*Musilova et al., 2019*). Since light with longer wavelengths is absorbed more quickly than those with shorter wavelengths (except for the shortest UV wavelengths), high-energy light with shorter wavelengths, such as blue, is able to penetrate to greater depths (*Figure 2A*). The genes responsible for absorbing these shorter wavelengths (*sws2* and *rh2*) are therefore much more important to deep-sea species in the photic zone (above 200 m) than those that absorb longer wavelengths (*lws*). Similarly, the genes providing monochromatic vision in very dim light (*rh1* and *gnat1*) have been proven to be important for deep-sea species of the disphotic zone (from 200 m to 1,000 m) (*Musilova et al., 2019*). We noticed that the *lws* gene (long wavelength) has been completely lost in both hadal snailfish and Tanaka's snailfish; *rh2* (central wavelength) has been specifically lost in hadal snailfish (*Figure 2B and C*); *sws2* (short wavelength) has undergone pseudogenization in hadal snailfish (*Figure 2—figure supplement 1*); while *rh1* and *gnat1* (perception of very dim light) is both still present and expressed in the eyes of hadal snailfish (*Figure 2D*). A previous study has also proven the existence of rhodopsin protein in the eyes of hadal snailfish using proteome data (*Yan et al., 2021*). The preservation and expression of genes for the perception of very dim light suggest that they are still subject to natural selection, at least in the recent past.

## Highly expressed auditory genes

Do hadal snailfish compensate for the lack of vision when perceiving the external environment? The genes associated with the olfactory and auditory systems were investigated using both comparative

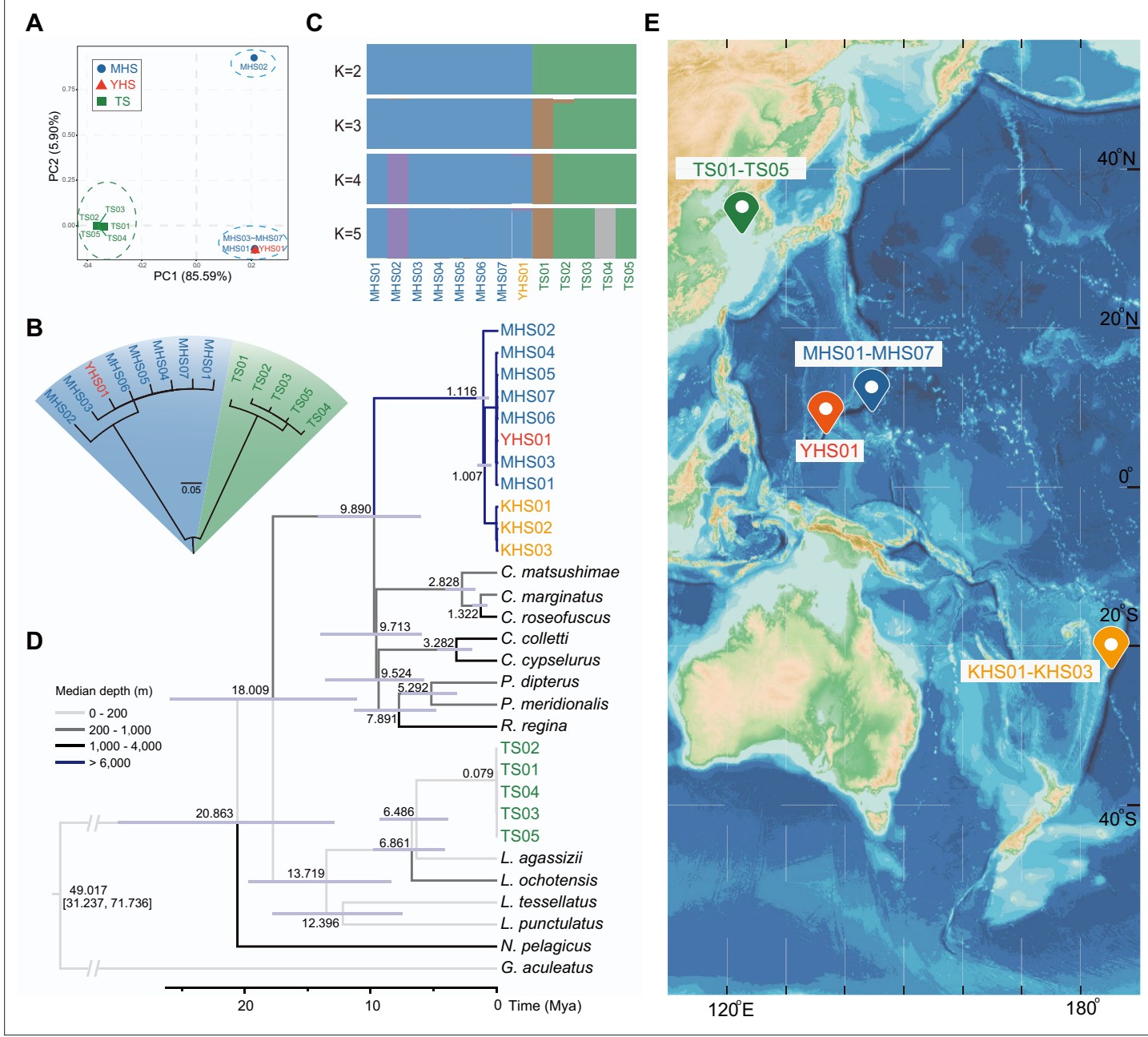

**Figure 1.** Sampling information of hadal snailfish, and phylogenetic relationships and population structure of resequenced individuals. (**A**) Principal component analysis (PCA) of eight hadal snailfish and five Tanaka's snailfish. PC, principal component; MHS, Mariana hadal snailfish; YHS, Yap hadal snailfish; TS, Tanaka's snailfish. (**B**) Neighbor-joining tree analysis of eight hadal snailfish and five Tanaka's snailfish using SNPs detected in whole-genome resequencing data. (**C**) Ancestry results from Admixture under the k = 5 model. (**D**) Maximum likelihood trees constructed with 13 genes encoding mitochondria in these species, where KHS01-KHS03 were constructed using two mitochondrial genes (*co1* and *cytb*) and manually merged with other species. Divergence times are shown in each node, and the color of each branch represents the survival depth of the species. KHS, *Notoliparis kermadecensis*. (**E**) Sampling information of hadal snailfish and Tanaka's snailfish. Blue represents the ocean, the darker the color, the deeper the depth, depth data from GEBCO Compilation Group (2020) GEBCO 2020 Grid (doi:10.5285/a29c5465-b138-234d-e053-6c86abc040b9).

The online version of this article includes the following figure supplement(s) for figure 1:

**Figure supplement 1.** K-mer (k = 27) distribution of the hadal snailfish (**A**) and Tanaka's snailfish (**B**).

**Figure supplement 2.** Genome assembly of hadal snailfish (**A**) and Tanaka's snailfish (**B**), both of them assembled 24 chromosomes.

**Figure supplement 3.** Improved genome assembly for hadal snailfish.

**Figure supplement 4.** Chromosomal syntenic relationship of hadal snailfish, Tanaka's snailfish, medaka, and stickleback.

*Figure 1 continued on next page*

*Figure 1 continued*

**Figure supplement 5.** The divergence time between Yap hadal snailfish (YHS) and Mariana hadal snailfish (MHS).

**Figure supplement 6.** Phylogenetic relationships of the family Liparidae.

**Figure supplement 7.** Diversity statistics.

**Figure supplement 8.** Demographic analysis.

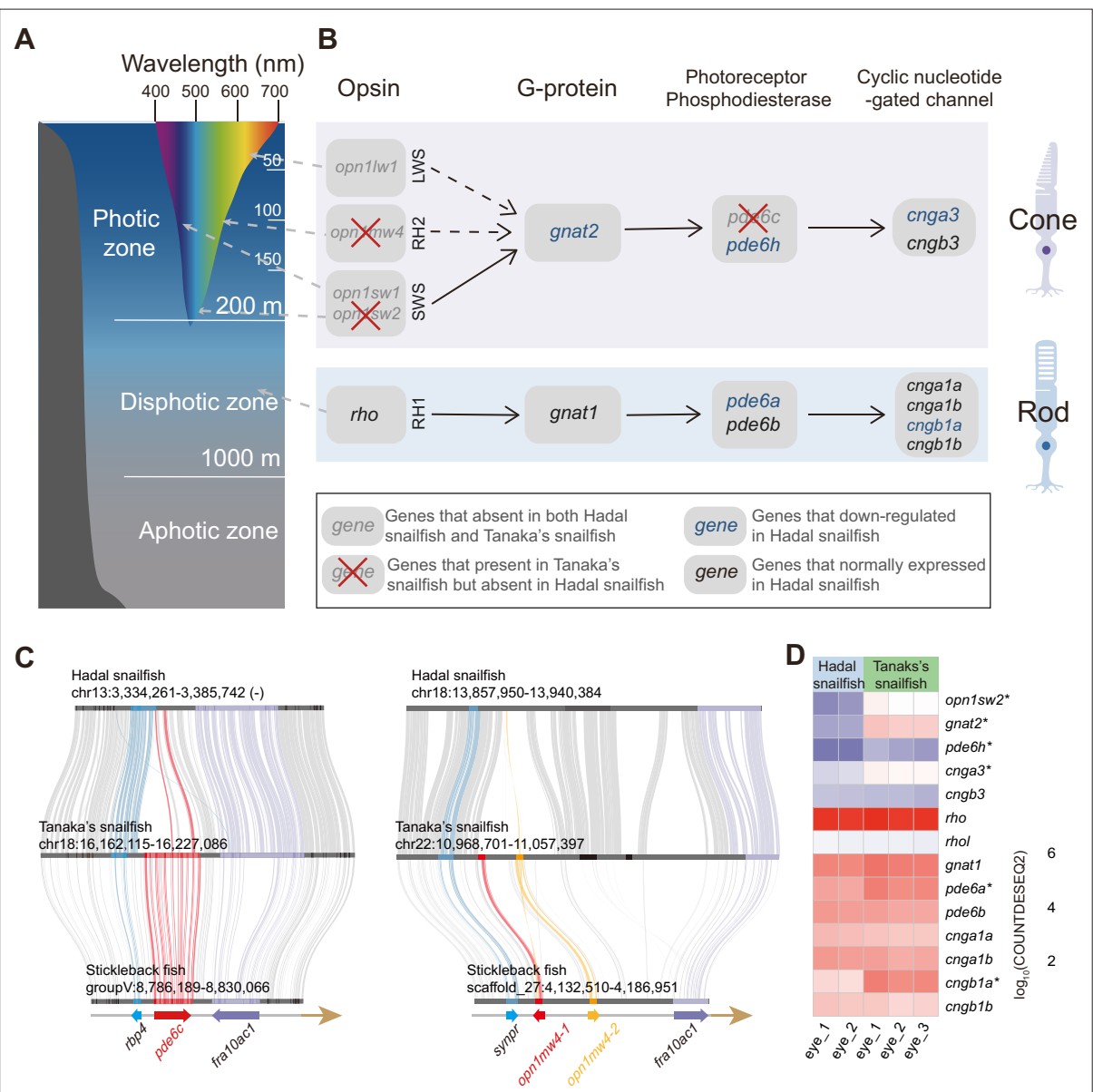

**Figure 2.** Alterations in vision-related genes in hadal snailfish. (**A**) Different colors of light penetrate the depth of the open ocean. Longer wavelengths (such as red) are absorbed at shallower depths, while shorter wavelengths (such as blue) can penetrate to deeper depths. (**B**) Genetic alterations in the genes encoding the four major proteins involved in activating the photoresponse of vertebrate photoreceptors in the cone cell and rod cell of hadal snailfish. Opsion: rhodopsin, or its cone equivalent. G-protein: heterotrimeric G-protein, transducin. (**C**) Gene loss of *pde6c* and *opn1mw4* in hadal snailfish. (**D**) Log10-transformation normalized counts for DESeq2 (COUNTDESEQ2) of vision-related genes in the eyes of hadal snailfish and Tanaka's snailfish. * represents genes significantly downregulated in hadal snailfish (corrected p<0.05).

The online version of this article includes the following figure supplement(s) for figure 2:

**Figure supplement 1.** Pseudogenization of *opn1sw2* in hadal snailfish.

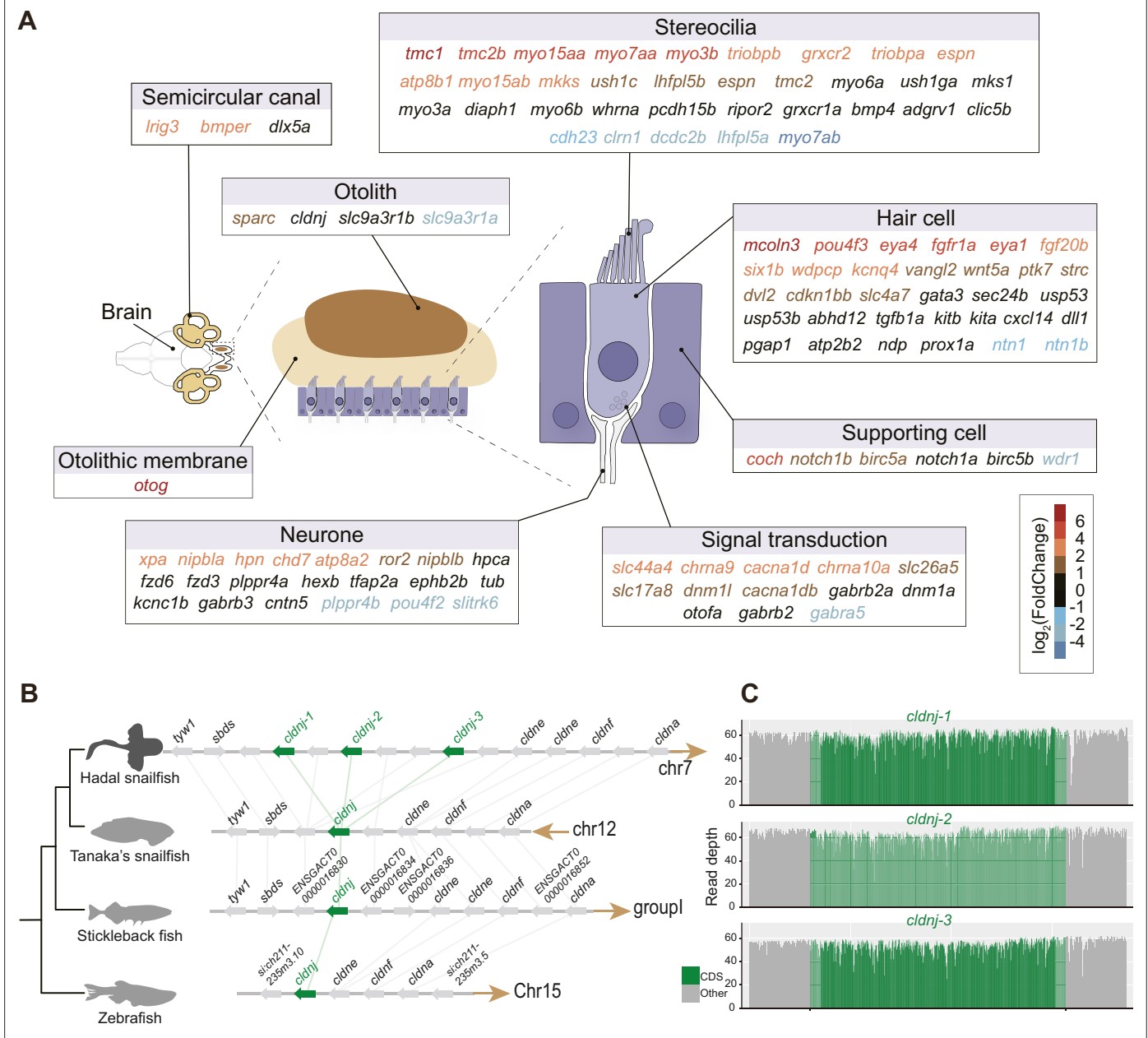

**Figure 3.** High expression and gene expansion of hearing-related genes in hadal snailfish. (**A**) Upregulation of auditory-related genes in hadal snailfish brain. Red represents upregulated genes in hadal snailfish. (**B**) Increased copy number of *cldnj* in hadal snailfish. The relative positions of genes on chromosomes are indicated by arrows, with arrows to the right representing the forward strand and arrows to the left representing the reverse strand. (**C**) Nanopore sequencing read depth for all *cldnj* in hadal snailfish.

The online version of this article includes the following figure supplement(s) for figure 3:

**Figure supplement 1.** The number of olfactory receptors in eight species.

**Figure supplement 2.** Specific changes of *tmc1* in hadal snailfish.

genomic and transcriptomic methods. While the number of olfactory receptors was largely reduced (***Figure 3—figure supplement 1***), we found that the majority of the auditory genes were well preserved in hadal snailfish. Many of the auditory genes also tended to be significantly more upregulated in the brain of hadal snailfish than in Tanaka's snailfish (***Figure 3A***; ***Supplementary file 14***). The upregulated genes involve many aspects of the auditory system, including the development and tethering of otoliths (***Kang et al., 2008***; ***Stooke-Vaughan et al., 2015***), the development (***Iyer and Groves, 2021***;

*Kozlowski et al., 2005*; *Riley, 2021*; *Wang et al., 2008*), maturation and maintenance of inner ear hair cells, the development and mechanosensitivity of stereocilia (*Cirilo et al., 2021*; *Kitajiri et al., 2010*), and other factors (*Giffen et al., 2019*; *Verdoodt et al., 2021*; *Figure 3A*). Of these, the most significant upregulated gene is *tmc1*, which encodes transmembrane channel-like protein 1, involved in the mechanotransduction process in sensory hair cells of the inner ear that facilitates the conversion of mechanical stimuli into electrical signals used for hearing and homeostasis (*Maeda et al., 2014*), and some mutations in this gene have been found to be associated with hearing loss (*Kitajiri et al., 2007*; *Riahi et al., 2014*). Interestingly, *tmc1* is also found to be the gene with the longest deletion specific to hadal snailfish (11 amino acids) in the regions that are generally highly conserved across vertebrate's genomes (*Figure 3—figure supplement 2*); the functional implications of this alteration need further verification.

Moreover, the gene involved in lifelong otolith mineralization, *cldnj*, has three copies in hadal snailfish, but only one copy in other teleost species, encoding a claudin protein that has a role in tight junctions through calcium-independent cell-adhesion activity (*Figure 3B and C*; *Hardison et al., 2005*). This may be important for hadal snailfish because calcium carbonate, the inorganic component in otoliths, is thought not to accumulate efficiently below the carbonate compensation depth (CCD; >4,000–5,000 m) (*Jamieson, 2015*). It should be noted that the hadal snailfish survive at depths far beyond the limits of CCD, but their otoliths still maintain densities similar to those of sea-surface species (*Gerringer et al., 2021a*). In our investigation, we found that the expression of *cldnj* was not significantly upregulated in the brain of the hadal snailfish than in Tanaka's snailfish, which may be related to the fact that *cldnj* is mainly expressed in the otocyst, while the expression in the brain is lower. However, due to the immense challenge in obtaining samples of hadal snailfish, the expression of *cldnj* in the otocyst deserves more in-depth study in the future. Expansion of *cldnj* was observed in all resequenced individuals of the hadal snailfish (*Supplementary file 10*), which provides an explanation for the hadal snailfish breaks the depth limitation on calcium carbonate deposition and becomes one of the few species of teleost in hadal zone.

## Circadian rhythm decoupled from sunlight and dark adaptation

There is growing evidence that persistent darkness challenges the physiology and behavior of animals, leading to disrupted circadian rhythms, neurological damage, and depressive-behavioral phenotypes (*Fisk et al., 2018*). Consistent with previous research in cavefish (*Policarpo et al., 2021*), we noticed that many of the circadian rhythm genes (*per2a, cry1a, cry3, cry5,* and *gpr19*) are lost or have undergone pseudogenization in the hadal snailfish (*Figure 4A and B*, *Figure 4—figure supplement 1*). Despite that, we noticed that the essential clock control genes are present and expressed in the hadal snailfish, indicating that the rhythm cycle is retained, although it is likely to have been largely uncoupled from sunlight. Moreover, *gpr19* deficiency has been reported to prolong the cycle of circadian locomotor activity rhythms (*Yamaguchi et al., 2021*), so hadal snailfish may have an extended rhythm cycle like cavefish (*Cavallari et al., 2011*; *Yamaguchi et al., 2021*).

In addition, in the teleosts closely related to hadal snailfish, there are usually two copies of *grpr* encoding the gastrin-releasing peptide receptor; we noticed that in hadal snailfish one of them is absent and the other is barely expressed in brain (*Figure 4C*), whereas a previous study found that the *grpr* gene in the mouse suprachiasmatic nucleus (SCN) did not fluctuate significantly during a 24 hr light/dark cycle and had a relatively stable expression (*Pembroke et al., 2015*; *Figure 4—figure supplement 1*). It has been reported that *grpr*-deficient mice, while exhibiting normal circadian rhythms, show significantly increased locomotor activity in dark conditions (*Wada et al., 1997*; *Zhao et al., 2023*). We might therefore speculate that the absence of that gene might in some way benefit the activity of hadal snailfish under complete darkness.

It should be noted that the abovementioned missing genes are not sufficient to exhibit the full range of changes that occur in the nervous system of hadal snailfish. Previous studies suggest that HHP suppressed the compound action potential in nerve trunks of fishes from shallow areas but not from deep areas, and can perturb the function of G protein-coupled receptors (*Siebenaller and Murray, 1995*). From our transcriptome data, we also observed that the brain is one of the most divergent organs regarding expression levels between hadal snailfish and Tanaka's snailfish (*Figure 4—figure supplement 2*). Specifically, there are 3,587 upregulated genes and 3,433 downregulated genes in the brain of hadal snailfish compared to Tanaka snailfish, and Gene Ontology (GO) functional enrichment

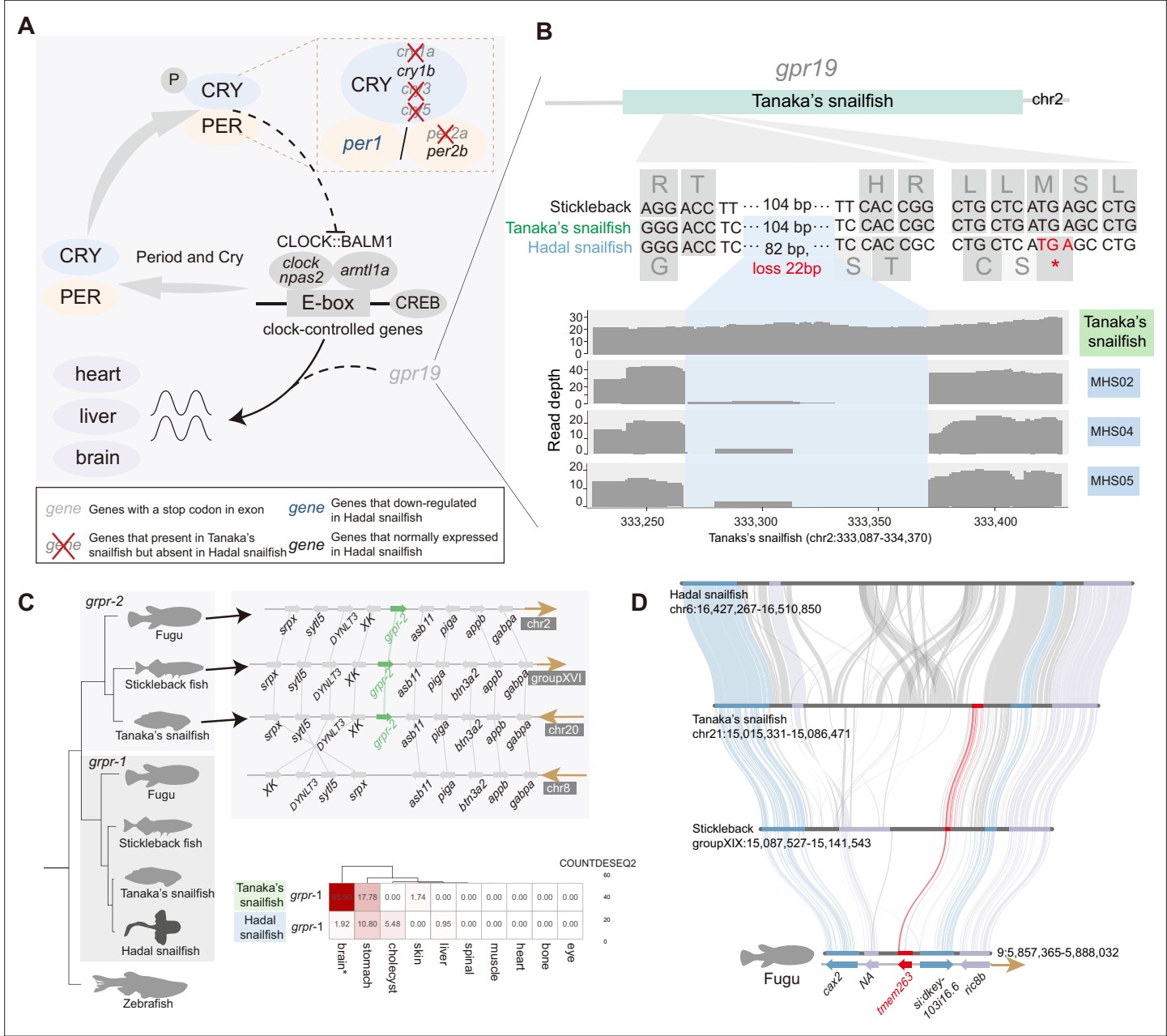

**Figure 4.** Genetic variation of dark adaptation in hadal snailfish. (**A**) Genetic changes involved in light-mediated regulation of the molecular clock in hadal snailfish suprachiasmatic nucleus (SCN) neurons. (**B**) Pseudogenization of *gpr19* (gray) in hadal snailfish. Gene structure (top), alignment of nucleotide and amino acid sequences (middle), and sequencing read depth (bottom; the numbers along the x-axis represent the position of the base at the scaffold) for the *gpr19* gene. The premature termination (colored red) of *gpr19* is due to 82 nucleotide variants and 22 nucleotide deletions (blue). (**C**) The deletion of one copy of *grpr* and another copy of downregulated expression in hadal snailfish. The relative positions of genes on chromosomes are indicated by arrows, with arrows to the right representing the forward strand and arrows to the left representing the reverse strand. The heatmap presented is the average of the normalized counts for DESeq2 (COUNTDESEQ2) in all replicate samples from each tissue. * represents tissue in which the *grpr-1* was significantly downregulated in hadal snailfish (corrected p<0.05). (**D**) Gene loss of *tmem263* in hadal snailfish.

The online version of this article includes the following figure supplement(s) for figure 4:

**Figure supplement 1.** Rhythm-related gene alterations in hadal snailfish.

**Figure supplement 2.** Tissue-specific changes in hadal snailfish.

**Figure supplement 3.** Pseudogenization of *gpr27* in hadal snailfish.

**Figure supplement 4.** Expression of the vitamin D-related genes in various tissues of hadal snailfish and Tanaka's snailfish.

**Figure supplement 5.** Loss of skeletal formation-related genes and site-specific mutations in hadal snailfish.

analyses revealed that upregulated genes in the hadal snailfish are associated with cilium, DNA repair, and microtubule-based movement, while downregulated genes are enriched in membranes, GTP-binding, proton transmembrane transport, and synaptic vesicles (*Supplementary file 15*). In line with this observation, one of our previous studies showed that zebrafish brains have the highest number of differentially expressed genes than the other investigated organs when exposed to HHP (*Hu et al., 2022*). We also identified 15 de novo new genes in hadal snailfish that are highly expressed in the brain (*Figure 4—figure supplement 2*). The adaptation of the nervous system to HHP deserves more in-depth study in the future.

## Possible survival strategy of storing energy

In a previous study, it was noticed that the individual hadal snailfish we investigated retained a large amount of intact food in its stomach and had larger eggs than might otherwise be expected (*Gerringer et al., 2017b*; *Wang et al., 2019*). It appears that the hadal snailfish have a survival strategy of storing energy, which is often found in species that need to cope with occasional starvation. Here we find another clue that hints at the existence of this possibility: the pseudogenization of the gene *gpr27* in hadal snailfish (*Figure 4—figure supplement 3*). *Gpr27* is a G protein-coupled receptor, belonging to the family of cell surface receptors, involved in various physiological processes and expressed in multiple tissues including the brain, heart, kidney, and immune system. It has been reported that the knockout of *gpr27* increases the expression of key enzymes in the carnitine shuttle complex (*Nath et al., 2020*), especially *cpt1*, which is essential for the β-oxidation of lipid metabolism. The transcriptome data further confirm that the gene *cpt1* is significantly upregulated in the liver of hadal snailfish. As lipid mobilization is thought to be a common metabolic response to short-term starvation in fish (*Liao et al., 2017*), the inactivation of *gpr27* could help hadal snailfish to better survive periods of food deficiency. Although previous surveys have shown that various types of organisms live in the hadal zone, and that the hadal snailfish can survive by eating amphipods and occasionally polychaetes and decapod shrimp (*Yan et al., 2021*), short-term starvation is still possible because although energy sources are limited by complete darkness, but organisms usually tend to 'over-reproduce.

## Reduced bone mineralization

Vitamin D synthesis is dependent on UV light, with phytoplankton being the origin of vitamin D in food (*Björn and Wang, 2000*). Whether and how vitamin D reaches the hadal zone through various pathways, for instance, as particulate organic matter, is still unknown. By investigating the genes associated with vitamin D metabolic pathways, we found that these genes are well conserved in the genome of hadal snailfish and are similarly expressed in both hadal snailfish and Tanaka's snailfish (*Figure 4—figure supplement 4*), suggesting that vitamin D may not be a limiting factor for hadal zone vertebrates.

Nonetheless, micro-CT scans have revealed shorter bones and reduced bone density in hadal snailfish, from which it has been inferred that this species has reduced bone mineralization (*Gerringer et al., 2021a*); this may be a result of lowering density by reducing bone mineralization, allowing to maintain neutral buoyancy without expending too much energy, or it may be a result of making its skeleton more flexible and malleable, which is able to better withstand the effects of HHP. The gene *bglap*, which encodes a highly abundant bone protein secreted by osteoblasts that binds calcium and hydroxyapatite and regulates bone remodeling and energy metabolism, had been found to be a pseudogene in hadal fish (*Wang et al., 2019*), which may contribute to this phenotype. Here, we found two more lost genes specific to hadal snailfish, *tmem251* and *tmem263*, that contribute to reduced bone mineralization (*Figure 4D*, *Figure 4—figure supplement 5*). These two genes encode transmembrane proteins, and loss-of-function mutations have now been found that may affect bone mineral deposition and thus bone development and body growth (*Ain et al., 2021*; *Wu et al., 2018*). Furthermore, many genes that determine chondrocyte differentiation and bone mineralization were found to be differentially expressed in the bones of hadal snailfish and Tanaka's snailfish. However, it should be noted that this result derives from a single bone sample of a hadal snailfish and needs further verification.

## HHP adaptation at cellular levels

HHP exerts broad effects upon cells, including cell membrane fluidity (*Casadei et al., 2002*; *Chong et al., 1983*; *Kato et al., 2002*), protein structure stability (*Abe, 2021*; *Gross and Jaenicke, 1994*), and oxidative stress (*Aertsen et al., 2005*; *Moserova et al., 2017*). In regard to the effect of cell membrane fluidity, relevant genetic alterations had been identified in previous studies, that is, the amplification of *acaa1* (encoding acetyl-CoA acetyltransferase 1, a key regulator of fatty acid β-oxidation in the peroxisome, which plays a controlling role in fatty acid elongation and degradation) may increase the ability to synthesize unsaturated fatty acids (*Fang et al., 2000*; *Wang et al., 2019*). As for the stability of the protein structure, previous studies have suggested that the high level of TMAO content could help the marine fishes in resistance to the inhibitory effects of high pressure on numerous proteins (*Ma et al., 2014*; *Yancey et al., 2002*; *Yancey et al., 2014*). We also observed another gene duplication event associated with protein stability. The gene *vbp1* (*Figure 5—figure supplement 1*; *Vainberg et al., 1998*), encoding prefoldin subunit 3 that promotes protein folding, has two copies in hadal snailfish but one copy in other teleost fishes. But unfortunately, although it is widely known that high pressure leads to the accumulation of reactive oxygen species (ROS) (*Abe, 2021*; *Aertsen et al., 2004*; *Le et al., 2020*), it is still unknown how deep-sea fish cope with this challenge.

We further examined the known ROS-related genes in hadal snailfish, but found that they were not significantly altered in sequence or expression (*Figure 5—figure supplement 1*). Next, we identified 34 genes that are significantly more highly expressed in all organs of hadal snailfish in comparison to Tanaka's snailfish and zebrafish, while only 7 genes were found to be significantly more highly expressed in Tanaka's snailfish using the same criterion (*Figure 5—figure supplement 1*). The 34 genes are enriched in only one GO category, GO:0000077: DNA damage checkpoint (adjusted p-value: 0.0177). Moreover, 5 of the 34 genes are associated with DNA repair. Interestingly, however, when we analyzed the genes that were both expanded and highly expressed in most tissues, we identified only one gene, *fthl27* (encoding a ferritin heavy chain-like protein), which has 14 copies (most of which are tandem duplicates) in hadal snailfish as opposed to 3 copies in Tanaka snailfish (*Figure 5A*, *Figure 5—figure supplement 2*). It has also been suggested that ferritin helps control ROS (*Orino et al., 2001*; *Salatino et al., 2019*). The expansion of *fthl27* was validated in all the eight resequencing individuals by reads mapping (*Figure 5—figure supplement 3*), indicating that the tandem duplication event occurred at least before the differentiation of these individuals. To test whether the *fthl27* can resist oxidative stress, we cultured 293T cells with or without *fthl27*-overexpression plasmid in cell culture medium supplemented with $H_2O_2$ or ferric ammonium citrate (FAC) for 4 hr, and subsequently measured intracellular ROS levels as well as cell viability. The results showed that the intracellular ROS levels of *fthl27*-overexpression cells were significantly lower than that of the control group (*Figure 5B*, *Figure 5—figure supplement 4*). Meanwhile, the *fthl27*-overexpression cells were also found to had significantly higher cell viability (*Figure 5C*). Therefore, we hypothesize that the expansion and high expression of this gene may be an important mechanism of resistance to HHP induced ROS in hadal snailfish.

## Discussion

The more sequenced individuals provide us with more details about the evolutionary history about the hadal snailfish. For example, given that the divergence time of the hadal snailfish and the other species of the family Liparidae living at a depth of 1,000 m was about 9.9 Mya, and the divergence time between different sequenced hadal snailfish individuals was about 1.1 Mya, it is known that the hadal snailfish entered the hadal zone between 1.1 and 9.9 Mya. Then consider the fact that the genes that are responsible for detecting light in dark environment are well preserved in the hadal snailfish, it is likely that this species have only entered a completely light-free environment in the last millions of years, after the full completion of the Mariana Trench (*Oakley et al., 2009*). In addition, the phylogenetic relationships between different individuals clearly indicate that they have successfully spread to different trenches within 1.0 Mya (*Figure 1—figure supplement 6*).

The comparative genomic analysis revealed that the complete absence of light had a profound effect on the hadal snailfish. In addition to the substantial loss of visual genes and loss of pigmentation, many rhythm-related genes were also absent, although some rhythm genes were still present.

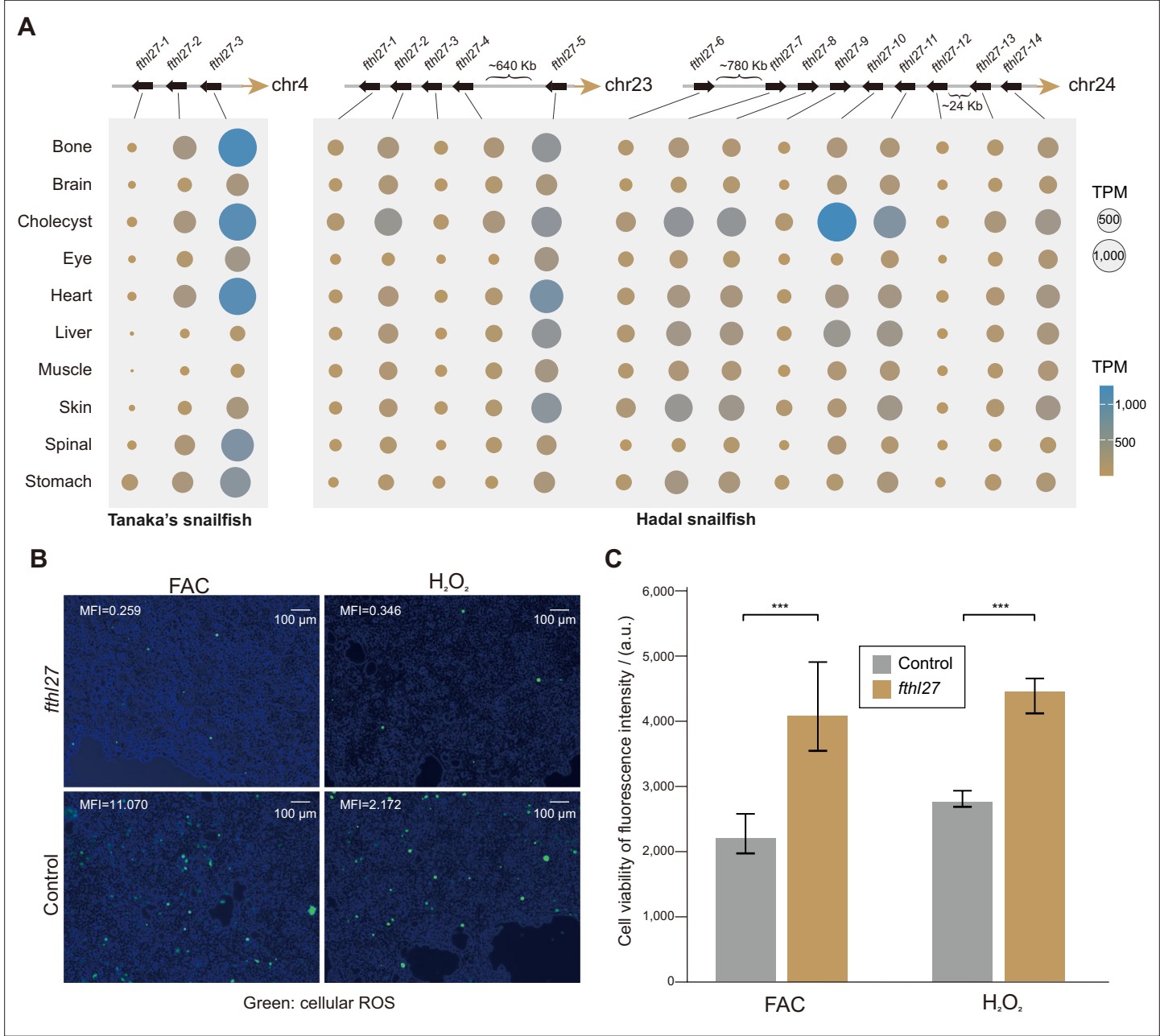

**Figure 5.** High-hydrostatic pressure adaptation of molecules and cells in hadal snailfish. (**A**) The position of the gene copies of *fthl27* in Tanaka's snailfish and hadal snailfish, and the expression of each gene copy in each tissue. The gene expression presented is the average of transcripts per million (TPM) in all replicate samples from each tissue. (**B**) Reactive oxygen species (ROS) levels were confirmed by redox-sensitive fluorescent probe using DCFH-DA molecular probe in 293T cell culture medium with or without *fthl27*-overexpression plasmid added with $H_2O_2$ or ferric ammonium citrate (FAC) for 4 hr. Images are merged from bright-field images with fluorescent images using ImageJ, while the mean fluorescence intensity (MFI) is also calculated using ImageJ. Green, cellular ROS. Scale bars = 100 μm. (**C**) Fluorescence intensities of AlamarBlue for 293T cells with or without the *fthl27*-overexpression plasmid in cell culture medium supplemented with $H_2O_2$ or FAC for 4 hr (n=5). Vertical coordinates are relative fluorescence units, and higher relative fluorescence units indicate stronger cell viability. Significantly higher AlamarBlue fluorescence for treated cells compared to control cells (p<0.001) is indicated by an asterisk (***).

The online version of this article includes the following figure supplement(s) for figure 5:

**Figure supplement 1.** Genetic mechanisms of adaptation to high hydrostatic pressure in hadal snailfish.

**Figure supplement 2.** Ranking of the expression of individual copies of *fthl27* gene in hadal snailfish and Tanaka's snailfish in various tissues showed that all copies of *fthl27* in hadal snailfish have high expression.

**Figure supplement 3.** Reads depth of copy number expansion of *fthl27* in eight hadal snailfish resequenced individuals.

*Figure 5 continued on next page*

*Figure 5 continued*
**Figure supplement 4.** Analysis of reactive oxygen species (ROS) intracellular amounts and *fthl27* antioxidant protein levels in 293T cells.
**Figure supplement 5.** Specific amino acid sites in the fmo3 protein sequence in hadal snailfish.

The gene loss may not only come from relaxation of natural selection, but also for better adaptation. For example, the *grpr* gene copies are absent or downregulated in hadal snailfish, which could in turn increase their activity in the dark, allowing them to survive better in the dark environment (*Wada et al., 1997*). The loss of *gpr27* may also increase the ability of lipid metabolism, which is essential for coping with short-term food deficiencies (*Nath et al., 2020*).

The most interesting question about the hadal snailfish is why this is currently one of the very few observed vertebrate species capable of surviving and reproducing at such depths. TMAO, which is able to maintain protein function under high pressure, is thought to be a limiting factor in determining the depth at which fish can survive (*Yancey et al., 2014*). Results from our previous analysis suggested that positive selection on *fmo3* may be important in promoting TMAO synthesis; but after an updated genome and rigorous FDR correction, we found that these genes had not undergone any particular or significant alteration in the amino acid sequences, although this does not exclude the possibility of a small number of amino acids (three species-specific mutations in all the five copies) having an effect on enzyme activity (*Figure 5—figure supplement 5*). Since the expression of all five copies of *fmo3* is similar among the various tissues of Tanaka's snailfish and hadal snailfish (*Supplementary file 16*), it seems more likely that the mechanism associated with TMAO degradation is altered in hadal snailfish, although we do not yet have any additional evidence to support this because the genes associated with TMAO degradation are still unclear.

However, the levels of TMAO are not sufficient for us to understand why only the hadal snailfish can tolerate such HHP since this substance is widely present in marine fishes. In contrast, the tandem duplication events of two genes may play a more critical role in the adaptation of the hadal snailfish. The first event is the tandem duplication of *cldnj*, a gene essential for otolith formation (*Figure 3B*; *Hardison et al., 2005*). The two more copies may help the hadal snailfish to maintain the densities of their otoliths far beyond the limits of CCD. Since the dissolution rate of calcium carbonate increases with higher pressure, the otoliths stability may be one of the reasons limiting fishes to dive to even deeper regions. The second event is the massive expansion and high expression of *fthl27*. Our cellular experiments proved that this gene could help cells to resist ROS burst and protect it from various damages caused by oxidative stress. Meanwhile, comparative transcriptomic analysis observed multiple genes associated with DNA repair are significantly more highly expressed in all tissues of hadal snailfish than in other fishes, which also coincides with the presence of oxidative stress (*Figure 5—figure supplement 1*).

In summary, we provide chromosome-level genomes of hadal snailfish and Tanaka's snailfish, as well as additional transcriptome and resequencing data. We report here further advances in our understanding of the origin, specific characteristics, and adaptive mechanisms of the hadal snailfish.

## Materials and methods
### Sample collection and identification
All the experiments in this study were conducted in accordance with the preapproved guidelines of the Ethics Committee of the Institute of Deep-Sea Science and Engineering, Chinese Academy of Sciences (Sanya, China). The hadal snailfish samples were collected form one site in the Mariana Trench (142°26′E, 11°07′N) at depth of 7,254 m using the deep-sea lander Tianya with a surfacing time of 3 hr (*Supplementary file 1*). These specimens were identified as conspecific with *P. swirei* by morphological observations. Tanaka's snailfish specimens were collected in the southern Yellow Sea in 2018 and identified as *L. tanakae* on the basis of morphological observations.

### Genome sequencing and assembly
Genomic DNA was extracted from the muscle of four hadal snailfish collected from the Mariana Trench and four Tanaka's snailfish collected from the southern Yellow Sea. We generated a total of 47.8 gigabases (Gb) of Nanopore reads, 148.6 Gb of BGI short reads, and 123.3 Gb of Hi-C reads for

hadal snailfish; and 39.0 Gb of Nanopore reads, 130.3 Gb of BGI short reads, and 99.5 Gb of Hi-C reads for Tanaka's snailfish.

The genome sizes of hadal snailfish and Tanaka's snailfish were estimated by *k*-mer distribution analysis (K = 27) of SOApec v2 (*Luo et al., 2012*) to be 633.2 Mb and 539.9 Mb, respectively. We then assembled the hadal snailfish and Tanaka's snailfish genomes based on the filtered Nanopore sequencing data using wtdbg2 v2.4.1 (*Ruan and Li, 2020*) based on default parameters, followed by two rounds of error correction using NextPolish v1.0 (*Hu et al., 2020*) based on the filtered BGI sequencing data, and finally assembled them into chromosomal versions using 3D-DNA v180114 (*Dudchenko et al., 2017*) based on Hi-C data. Finally, BUSCO v 4.1.2 (*Manni et al., 2021*) was used with the library 'actinopterygii_odb10' to analyze and evaluate the completeness of the gene set in our draft genome.

## Transcriptome sequencing

A total of 11 transcriptomes from 6 tissues (eye, stomach, heart, liver, muscle, skin) were extracted from three hadal snailfish, while a total of 26 transcriptomes from 10 tissues (brain, spinal cord, eye, bone, cholecyst, stomach, heart, liver, muscle, skin) were extracted from three Tanaka's snailfish. RNA was subsequently extracted using TRIzol (Invitrogen) and purified using the RNeasy Mini Kit (QIAGEN). Transcriptome reads were obtained from the Illumina HiSeq 2000 sequencing platform. The RNA sequences were filtered using Fastp v0.20 (*Chen et al., 2018*) and assembled without reference using SPAdes (*Bushmanova et al., 2019*) with default parameters. Subsequently, TransDecoder(RRID:SCR_017647) v5.5.0 was used to identify coding regions of the transcripts.

## Genome annotation

Both de novo and homology-based predictions were used to identify repetitive elements in hadal snailfish and Tanaka's snailfish. First, we constructed a de novo transposable element library using RepeatModeler v1.0.11 (*Saha et al., 2008*), and then used RepeatMasker v4.0.7 (*Chen, 2004*) to detect repeats. For homologous annotations, the genome sequences were compared with data from Repbase using RepeatMasker v4.0.7 and RepeatProteinMask v1.36 to predict transposable elements. For tandem repeat sequences, we used Tandem Repeats Finder v4.07 (*Benson, 1999*) to make predictions.

The repeat masked genome was used for the gene annotation. We used a combination of ab initio gene predictions, homologous gene predictions, and direct gene models produced by transcriptome assembly to identify protein-coding genes structure on the genome as follows:

> Step 1: Augustus v3.2.1 (*Stanke et al., 2008*) was used to generate ab initio predictions with internal gene models.
> Step 2: The protein sequences from seven species – medaka, Atlantic cod, flatfish, stickleback, zebrafish, turbot, and fugu – and the transcriptome-predicted protein sequences were used to align genomic sequences with BLAT v. 35 (*Supplementary file 6*; *Kent, 2002*).
> Step 3: The psl files obtained in the previous step were integrated and the protein sequences that were aligned to the overlapping region of the genome were scored and sorted based on the alignment results using a custom script to filter out the best aligned protein sequences in this region. Then, GeneWise v2.4.1 (*Birney et al., 2004*) was used to predict gene models with the aligned sequences as well as the corresponding query proteins. The custom scripts have been deposited in GitHub (https://github.com/wk8910/bio_tools/tree/master/42.prediction copy archived at *Wang, 2021*).
> Step 4: The Evidence Modeler (EVM) v1.1.1 (*Haas et al., 2008*) was used to integrate the prediction results with different weights for each.

The integrated gene set was translated into amino acid sequences using InterProScan v5 (*Jones et al., 2014*) to annotate motifs and domains in protein sequences by searching publicly available databases (including Pfam, PRINTS, PANTHER, ProDom, and SMART), and the genes were further annotated using the KEGG databases.

## Variant calling using resequencing data

Short reads of seven Mariana hadal snailfish, one Yap hadal snailfish, and five Tanaka's snailfish (*Supplementary file 1*) were mapped to the hadal snailfish genome assembled in this study with

BWA v0.7.12-r1039 (*Li, 2013*); then SAMtools v1.4 (*Li et al., 2009*) was used to sort and obtain BAM files. To analyze population genetics, we focused on SNPs and small indels (1–10 bp) (*Zhang et al., 2021*). The SNPs were called using FreeBayes v0.9.10-3-g47a713e (*Garrison and Marth, 2012*) with parameters '`--gvcf --min-coverage 5 --limit-coverage 200`' and filtered by following three thresholds: (1) SNPs with missing rate ≤30%; (2) the highest sequencing depth of SNP position <200×; and (3) the lowest sequencing depth for each allele ≥5. Subsequently, we calculated the distribution of heterozygosity in genomewide regions with 500 kb nonoverlapping sliding windows.

## Inference of phylogeny history

### SNP tree, PCA, and diversity statistics

PLINK v1.90b6.6 (*Chen et al., 2019*) was used to perform PCA and other population divergency statistics, including nucleotide diversity and genetic differentiation (FST). A neighbor-joining tree was constructed with PHYLIP v3.697 (*Felsenstein, 1993*) for paired genetic distance matrices.

### Admixture analysis

Different K values (from 1 to 5) were tested using Admixture v1.3.0 (*Alexander et al., 2009*) to infer ancestral populations in all hadal snailfish and Tanaka's snailfish individuals accessions.

### Demographic analysis

The demographic history of hadal snailfish and Tanaka's snailfish was inferred with pairwise sequential Markovian coalescent (PSMC) (*Li and Durbin, 2011*) analysis, based on a substitution rate of 1.9174e-09 per generation for hadal snailfish and 5.6790e-09 per generation for Tanaka's snailfish. The analysis was performed using the following parameters: −N25 −t15 −r5 −p '4+25 × 2+4 + 6'. These mutation rates were estimated using r8s v1.81. The generation time is 1 y for Tanaka snailfish and 3 y for hadal snailfish.

### Mitochondrial genome phylogenetic reconstruction and divergence time estimation

The mitochondria of eight hadal snailfish and five Tanaka's snailfish were assembled using NOVOPlasty v4.3.1 (*Dierckxsens et al., 2017*) with default parameters and annotated using MITOS (http://mitos2.bioinf.uni-leipzig.de/index.py). Subsequently, mitochondrial data from currently published species of the Liparidae were combined, and nucleic acid sequences of 13 coding genes on mitochondria were aligned with MUSCLE v3.8.425 (*Edgar, 2021*) using default parameters, and alignments of the coding sequences were generated with pal2nal v14 using default parameters. The maximum likelihood (ML) tree was constructed with RAxML-8.2.12 (*Stamatakis, 2014*) using the following parameters: -f a -m GTRGAMMA -p 15256 -x 271828 -N 100. Finally, divergence times were estimated using MCMCtree v4.9j (*Yang, 2007*) with one soft-bound calibration timepoint (snailfish-stickleback: ~32–73 Ma) based on previous studies. For *Notoliparis kermadecensis*, we combined all the above mitochondrial data and performed the same above analysis based on *co1* and *cytb* gene sequences to obtain the ML tree and divergence times.

## Gene loss and duplication

Here, we applied an improved read mapping-based method to identify gene loss and duplication, which is effective in reducing false positives and false negatives caused by genome assembly and annotation errors as well as multispecies sequence alignments. The custom scripts have been deposited in GitHub (https://github.com/wenjie-xu-nwpu/hadal_snailfish copy archived at *Xu, 2023*). Although this method may have limitations for identifying gene loss and duplication in species with long divergence times, the divergence times of hadal snailfish and Tanaka's snailfish are about 20 million years (*Wang et al., 2019*), and at least 88% of the reads in all hadal snailfish individuals can be well compared to Tanaka's snailfish genome, indicating that this method is applicable to this study.

For gene loss, the following methods were used for identification. (1) Short reads of eight hadal snailfish and five Tanaka's snailfish (~30×) were compared to the stickleback and Tanaka's snailfish genome using BWA v0.7.12-r1039 (*Li, 2013*) and subsequently sorted using SAMtools v1.4 (*Li et al., 2009*) to obtain the BAM files. (2) We obtained the reads depth for each sites in the gene

coding region based on the annotation information of the reference genome and subsequently classified the depths we had on individual loci into three types ('HIGH' for greater than half of the average coverage, 'LOW' for less than 3, and 'MID' for the rest). We defined sites with 'HIGH' for Tanaka's snailfish and 'LOW' for hadal snailfish as hadal snailfish-specific lost sites (SLSs). Then, the genes with SLSs accounting for at least 40% of the coding sequence length were selected as the candidate specific loss genes. (3) The protein sequences of the genes selected in the previous step were used as a reference to search through the genome of hadal snailfish using BLAT v. 35 (*Kent, 2002*) and predict the gene structure using GeneWise v2.4.1 (*Birney et al., 2004*) to determine the genes that were completely lost or partially lost in this species. (4) The synteny alignment between the hadal snailfish, Tanaka's snailfish, and stickleback was plotted for partial or fully lost of the gene.

For gene duplication, the following methods were used for identification. (1) Short reads of eight hadal snailfish and five Tanaka's snailfish were compared to the stickleback and Tanaka's snailfish genome using BWA v0.7.12-r1039 (*Li, 2013*) and subsequently sorted using SAMtools v1.4 (*Li et al., 2009*) to obtain the BAM files. (2) The homologous sites whose average value of reads depth of all hadal snailfish individuals were greater than 1.5 the average value of the Tanaka's snailfish individuals were retained and defined as hadal snailfish specific high-copy sites (HCSs). Then, the genes with HCSs accounting for at least 50% of the coding sequence length were selected as the candidate high-copy genes. (3) We searched for the location of this gene on the hadal snailfish genome using BLAT v. 35 (*Kent, 2002*) and predicted the gene structure using GeneWise v2.4.1 (*Birney et al., 2004*) to determine its copy number. (4) Finally, the expansion of this gene was determined by constructing a gene tree of the protein sequences of this gene family from nine species: hadal snailfish, Tanaka's snailfish, medaka, Atlantic cod, flatfish, stickleback, zebrafish, turbot, and fugu.

## Identification of unitary pseudogenes

Unitary pseudogenes are nonfunctional genes that decay at their original location (*Tutar, 2012*), and we suggest that some missing homologs will exist in hadal snailfish genome as unitary pseudogenes during their adaptation to the special environment of the hadal zone.

We obtained pseudogenes in hadal snailfish by following five steps. (1) Using the stickleback genome sequence as a reference, we performed synteny alignment for three species (hadal snailfish, Tanaka's snailfish, and stickleback) with Last v956 (*Kiełbasa et al., 2011*) using the parameters '-E 0.05', generating a total of 382 Mb (of which 290 Mb was informative for all species) of one-to-one alignment sequences with Multiz v1 (*Blanchette et al., 2004*) using the default parameters. (2) Genes with at least 70% of the coding sequences of stickleback or Tanaka's snailfish present in the MAF and not present in the corresponding regions of hadal snailfish were selected as alternative unitary pseudogene datasets. (3) We used BLAST v2.9.0 (*Altschul et al., 1990*) to determine if this gene was present in other regions of the hadal snailfish genome. (4) The hadal snailfish corresponding region was extended left and right by 10 kb, and the genes of stickleback and Tanaka's snailfish were used as references for predict the gene structure using GeneWise v2.4.1 (*Birney et al., 2004*). (5) Screening for pseudogenes that were consistent in all hadal snailfish individuals.

## De novo-originated new genes

First, the short reads of eight hadal snailfish and five Tanaka's snailfish were compared to the hadal snailfish genome using BWA v0.7.12-r1039 (*Li, 2013*) and subsequently sorted using SAMtools v1.4 (*Li et al., 2009*) to obtain the BAM files. In the second step, we defined a single-sequenced sample with reads depths <10 at a single locus as a deletion locus. Based on the annotation file of hadal snailfish, we screened all Tanaka's snailfish individuals for genes with deletions >50%. Next, for the genes specifically present in hadal snailfish selected in the previous step, we used BLAST v2.9.0 (*Altschul et al., 1990*) to align them with the genomes of eight other fishes (Tanaka's snailfish, medaka, Atlantic cod, flatfish, stickleback, zebrafish, turbot, and fugu) and screened for genes with a matching region <0.4. Genes with transcripts per million (TPM) maxima less than 1 in each tissue of hadal snailfish were filtered out. The fully annotated genes (presence of start and stop codons) in the results were defined as novel genes of hadal snailfish.

## Lineage-specific changes in amino acid sequences

For 17 species – Tanaka's snailfish, stickleback, pacific bluefin tuna, medaka, platy fish, Atlantic cod, flatfish, zebrafish, turbot, fugu, spotted gar, coelacanth, chicken, mouse, human, brownbanded bamboo shark, and elephant shark (*Supplementary file 6*) – we identified one-to-one orthologs for each species and hadal snailfish by the Reciprocal Best-Hits (RBH) method, and subsequently selected genes present in 15 species, including hadal snailfish, as conserved gene sets. Next, the protein sequences of the selected genes were aligned using MAFFT v7.471 (*Katoh and Standley, 2013*), and a custom script was used to select regions that were consistent in other species and had contiguous specificity at sites greater than 3 bp in hadal snailfish, and that had at least 90% sequence identity for each 5 bp region before and after this variant region (*Wu et al., 2021*). Finally, genes with consistent variants in all hadal snailfish individuals were selected.

We performed protein structure simulation using AlphaFold2 (*Cramer, 2021*) for the amino acid sequences of target genes in hadal snailfish and Tanaka's snailfish. Finally, the highest scoring prediction was selected as the best structure and visualized using UCSF Chimera (*Pettersen et al., 2004*).

## Comparative transcriptome analysis

For the RNA sequences of hadal snailfish and Tanaka's snailfish, we used Fastp v0.20.0 (*Chen et al., 2018*) to filter out low-quality and contaminated reads, and then used Hisat2 v 2.1.0 (*Kim et al., 2019*) to align them to the respective reference genomes. StringTie v1.3.6 (*Pertea et al., 2016*) was then used to count the number of reads paired for each gene with the help of gene annotation information of the species, and then TPM values were calculated for each gene in both species. Next, we identified 17,281 one-to-one orthologs of hadal snailfish and Tanaka's snailfish using the RBH method. Subsequently, we identified the genes that were differentially expressed (DEGs) between the same tissues of two species using the R package DESeq2 with $|\log_2$ (foldchange)$| \geq 1$ and corrected $p<0.05$. For genes that are upregulated or downregulated in multiple tissues, we first found by stochastic simulation that a gene is differentially expressed between two species in one organ does not affect the probability that this gene is differentially expressed in any other organ. Subsequently, we counted the genes that were upregulated or downregulated in each tissue to obtain a list of genes that were co-altered in multiple tissues.

## Cell lines

We selected human embryonic kidney (HEK) 293T cells as an in vitro model. HEK293T cells were provided by Fourth Military Medical University (Xi'an, China). The cell line was validated by short tandem repeat analysis and validated as negative for mycoplasma. The cells were maintained in DMEM (Gibco, USA) supplemented with 10% FBS (Gibco) and 1% antibiotic antifungal (Gibco) at 37°C, 5% $CO_2$.

## Cell culture, transfection, and ROS detection

HEK293T cells were inoculated in 6-well plates at a density of $4.0 \times 10^5$ cells/well. After a day when the cell density reached 50–60%, pcDNA3.1 and pcDNA3.1-*fthl27* were transfected into cells with polyethylenimine (PEI), respectively. After 24 hr of transfection, 6-well plates (2 ml total volume of medium per well) were treated with 200 µl of 0.3 mM $H_2O_2$ or 0.4 mM FAC, both for 4 hr. Subsequently, the ROS Reactive Oxygen Species Kit (Aladdin R272916-1000T) was used to measure ROS. ROS levels were measured using a DCFH-DA molecular probe, and fluorescence was observed through a fluorescence microscope with an optional FITC filter, with the background removed to observe changes in fluorescence. Bright-field photos and fluorescent photos were merged using ImageJ (RRID:SCR_003070) v1.53t. Subsequently, we added AlamarBlue reagent to the cells in complete medium, incubated them for 2 hr, and then assayed them with a fluorescence zymograph with excitation light wavelength between 530 and 560 nm and emission light wavelength of 590 nm, and recorded the relative fluorescence units.

## Acknowledgements

The project was supported by the National Key R&D Program of China (2022YFC3400300); the National Natural Science Foundation of China (32122021 to KW and 41876179 to SH); the 1000 Talent Project of Shaanxi Province to QQ, KW, and SH; Fundamental Research Funds of Northwestern Polytechnic University; Strategic Priority Research Program of Chinese Academy of Sciences (grant no. XDB42000000), Open Foundation from Marine Sciences in the First-Class Subjects of Zhejiang (No.OFMS011). The authors thank Dr. Yang Zhou, Dr. Yuan Yuan, and Dr. Tao Qin for their advice and discussions during this project.

## Additional information

### Funding

| Funder | Grant reference number | Author |
|---|---|---|
| National Key Research and Development Program of China | | Kun Wang |
| National Natural Science Foundation of China | | Shunping He |
| the 1000 Talent Project Shaanxi Province | | Qiang Qiu |
| Fundamental Research Funds of Northwestern Polytechnic University | | Kun Wang |
| Strategic Priority Research Program of Chinese Academy of Sciences | | Shunping He |

The funders had no role in study design, data collection and interpretation, or the decision to submit the work for publication.

### Author contributions

Wenjie Xu, Formal analysis, Visualization, Writing – original draft, Writing – review and editing; Chenglong Zhu, Chenguang Feng, Formal analysis, Visualization, Writing – original draft; Xueli Gao, Formal analysis, Validation, Methodology, X.G. performed experiments with fthl27; Baosheng Wu, Han Xu, Resources, Investigation; Mingliang Hu, Jiangmin Zheng, Formal analysis, Visualization; Honghui Zeng, Xiaoni Gan, Jing Bo, Li-Sheng He, Resources; Qiang Qiu, Funding acquisition, Writing – original draft, Writing – review and editing; Wen Wang, Writing – original draft, Writing – review and editing; Shunping He, Supervision, Funding acquisition, Writing – original draft, Writing – review and editing; Kun Wang, Supervision, Funding acquisition, Visualization, Writing – original draft, Project administration, Writing – review and editing

### Author ORCIDs

Wenjie Xu http://orcid.org/0000-0001-6240-8472
Mingliang Hu http://orcid.org/0000-0001-9018-6715
Kun Wang http://orcid.org/0000-0001-6059-6529

Reviewer #1 (Public Review): https://doi.org/10.7554/eLife.87198.3.sa1
Reviewer #2 (Public Review): https://doi.org/10.7554/eLife.87198.3.sa2
Author Response https://doi.org/10.7554/eLife.87198.3.sa3

## Additional files

### Supplementary files

• Supplementary file 1. Details of the samples used in this study.

- Supplementary file 2. Sequencing reads of hadal snailfish and Tanaka's snailfish.
- Supplementary file 3. Genome assembly statistics for hadal snailfish and Tanaka's snailfish.
- Supplementary file 4. Results of BUSCO evaluations of hadal snailfish and Tanaka's snailfish.
- Supplementary file 5. Transcriptome for hadal snailfish, Tanaka's snailfish, and zebrafish.
- Supplementary file 6. The genome information of the species used in this work.
- Supplementary file 7. Summary of repetitive sequences in hadal snailfish and Tanaka's snailfish.
- Supplementary file 8. The genes specifically lost in hadal snailfish.
- Supplementary file 9. The unitary pseudogenes identified in hadal snailfish.
- Supplementary file 10. List of genes with more copies in hadal snailfish compared to Tanaka's snailfish.
- Supplementary file 11. The genes with site-specific amino acid mutations (with a length ≥3 amino acids) in hadal snailfish.
- Supplementary file 12. Expression of 33 de novo-originated new genes in hadal snailfish in various tissues.
- Supplementary file 13. Depth of survival of some fishes of the Liparidae.
- Supplementary file 14. Expression of auditory-related genes in hadal snailfish.
- Supplementary file 15. GO enrichment of expression upregulated and downregulated genes in hadal snailfish brain.
- Supplementary file 16. The TPM of *fmo3* gene copies in hadal snailfish and Tanaka's snailfish.
- MDAR checklist

## Data availability

The sequence data files of the hadal snailfish have been deposited in the NCBI BioProject database with accession numbers PRJNA852951 (genome data) and PRJNA855356 (transcriptome data). The genome assembly file is under accession number JANBZZ000000000.

The following datasets were generated:

| Author(s) | Year | Dataset title | Dataset URL | Database and Identifier |
|---|---|---|---|---|
| Northwestern Polytechnical University | 2023 | Transcriptome sequencing data of Pseudoliparis swirei | https://www.ncbi.nlm.nih.gov/bioproject/PRJNA855356/ | NCBI BioProject, PRJNA855356 |
| Northwestern Polytechnical University | 2023 | Pseudoliparis swirei isolate:HS2019 Genome sequencing and assembly | https://www.ncbi.nlm.nih.gov/bioproject/PRJNA852951 | NCBI BioProject, PRJNA852951 |

The following previously published datasets were used:

| Author(s) | Year | Dataset title | Dataset URL | Database and Identifier |
|---|---|---|---|---|
| Northwestern Polytechnical University | 2018 | Pseudoliparis amblystomopsis Raw sequence reads | https://www.ncbi.nlm.nih.gov/bioproject/PRJNA472845/ | NCBI BioProject, PRJNA472845 |
| BGI-SZ | 2019 | Pseudoliparis sp. Yap Trench Genome sequencing and assembly | https://www.ncbi.nlm.nih.gov/bioproject/?term=PRJNA512070 | NCBI BioProject, PRJNA512070 |
| Genome Reference Consortium | 2019 | Stickleback assembly and gene annotation | https://ftp.ensembl.org/pub/release-97/fasta/gasterosteus_aculeatus/ | Ensembl, release-97/fasta/gasterosteus_aculeatus/ |
| Genome Reference Consortium | 2019 | Zebrafish assembly and gene annotation | https://ftp.ensembl.org/pub/release-97/fasta/danio_rerio/ | Ensembl, release-97/fasta/danio_rerio/ |

*Continued on next page*

*Continued*

| Author(s) | Year | Dataset title | Dataset URL | Database and Identifier |
|---|---|---|---|---|
| Genome Reference Consortium | 2019 | Spotted gar assembly and gene annotation | https://ftp.ensembl.org/pub/release-97/fasta/lepisosteus_oculatus/ | Ensembl, release-97/fasta/lepisosteus_oculatus/ |
| Genome Reference Consortium | 2019 | Mouse assembly and gene annotation | https://ftp.ensembl.org/pub/release-97/fasta/mus_musculus/ | Ensembl, GRCm39 |
| Genome Reference Consortium | 2019 | Human assembly and gene annotation | https://ftp.ensembl.org/pub/release-97/fasta/homo_sapiens/ | Ensembl, release-97/fasta/homo_sapiens/ |
| Genome Reference Consortium | 2019 | Medaka assembly and gene annotation | https://ftp.ensembl.org/pub/release-97/fasta/oryzias_latipes/ | Ensembl, release-97/fasta/oryzias_latipes/ |
| Genome Reference Consortium | 2019 | Platyfish assembly and gene annotation | https://ftp.ensembl.org/pub/release-97/fasta/xiphophorus_maculatus/ | Ensembl, release-97/fasta/xiphophorus_maculatus/ |
| Genome Reference Consortium | 2019 | Atlantic cod assembly and gene annotation | https://ftp.ensembl.org/pub/release-97/fasta/gadus_morhua/ | Ensembl, release-97/fasta/gadus_morhua/ |
| Genome Reference Consortium | 2019 | Turbot assembly and gene annotation | https://ftp.ensembl.org/pub/release-97/fasta/scophthalmus_maximus/ | Ensembl, release-97/fasta/scophthalmus_maximus/ |
| Genome Reference Consortium | 2019 | Fugu assembly and gene annotation | https://ftp.ensembl.org/pub/release-97/fasta/takifugu_rubripes/ | Ensembl, release-97/fasta/takifugu_rubripes/ |
| Genome Reference Consortium | 2019 | Coelacanth assembly and gene annotation | https://ftp.ensembl.org/pub/release-97/fasta/latimeria_chalumnae/ | Ensembl, release-97/fasta/latimeria_chalumnae/ |
| Genome Reference Consortium | 2019 | Chicken assembly and gene annotation | https://ftp.ensembl.org/pub/release-97/fasta/gallus_gallus/ | Ensembl, release-97/fasta/gallus_gallus/ |
| Genome Reference Consortium | 2019 | Elephant shark assembly and gene annotation | https://ftp.ensembl.org/pub/release-97/fasta/callorhinchus_milii/ | Ensembl, release-97/fasta/callorhinchus_milii/ |
| Fisheries Resources Institute, Japan Fisheries Research, Education Agency | 2021 | Genome assembly Tori_3.0 | https://www.ncbi.nlm.nih.gov/assembly/GCA_021601225.1/ | NCBI Assembly, GCA_021601225.1 |
| Tsinghua University | 2017 | Genome assembly Flounder_ref_guided_V1.0 | https://www.ncbi.nlm.nih.gov/assembly/GCF_001970005.1/ | NCBI Assembly, GCF_001970005.1 |
| Phyloinformatics Unit | 2018 | Genome assembly Cpunctatum_v1.0 | https://www.ncbi.nlm.nih.gov/assembly/GCA_003427335.1/ | NCBI Assembly, GCA_003427335.1 |

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
