## [Editor Report · eLife assessment]

This **important** study advances our understanding of the potential mechanisms of deep-sea adaptation and sheds light on the evolutionary history of hadal snailfish. Through comparative genomic analysis, the authors provide **convincing** evidence and propose hypotheses on the timing of trench colonization, population structure, and adaptations to the hadal snailfish genome in response to their environment.

---

## [Referee Report · Reviewer #1 (Public Review)]

This manuscript provides an important case study for in-depth research on the adaptability of vertebrates in deep-sea environments. Through analysis of the genomic data of the hadal snailfish, the authors found that this species may have entered and fully adapted to extreme environments only in the last few million years. Additionally, the study revealed the adaptive features of hadal snailfish in terms of perceptions, circadian rhythms and metabolisms, and the role of ferritin in high-hydrostatic pressure adaptation. Besides, the reads mapping method used to identify events such as gene loss and duplication avoids false positives caused by genome assembly and annotation. This ensures the reliability of the results presented in this manuscript. Overall, these findings provide important clues for a better understanding of deep-sea ecosystems and vertebrate evolution.

---

## [Referee Report · Reviewer #2 (Public Review)]

This paper presents improved, chromosome level assemblies of the hadal snailfish and Tanaka's snailfish. This is an extension and update of previous work from the group on the hadal snailfish genome. The chromosomal assemblies allow comparisons of genome architecture between a shallow water snailfish and the hadal snailfish to aid inference on timing of colonization of trenches and genomic changes that may have been adaptive for that move.

The comparisons in genomic architecture are compelling: genes present in Tanaka's snailfish that are lost in hadal snailfish that involve whole regions of the genome that no longer map even though adjacent regions do map between the species and across a large evolutionary distance to stickleback. Or genes that are duplicated in hadal snailfish but only appear as single copy in other fishes. The paper focuses on genes in the eye, in hearing, in circadian rythms, and in ROS scavaging. These are all functions that could play a role in adapting to the hadal environment.

The genomic comparisons all seem sound. Stylistically I would prefer if the authors could introduce the gene product and protein function every time they introduce a gene locus. They introduce a gene and general function, but don't usually note what the protein encoded by the gene is and what it's specific function is.

I found the paper generally well written, and the data compelling and creatively displayed.

Upon revision, the authors have commendably addressed all reviewer comments and added a slew of additional analyses. I find the paper stronger, better argued and have no further questions or comments.

---

## [Author Response]

The following is the authors’ response to the original reviews.

We would like to thank the reviewers for their thoughtful comments and constructive suggestions. Point-by-point responses to comments are given below:

**Reviewer #1 (Recommendations For The Authors):**
This manuscript provides an important case study for in-depth research on the adaptability of vertebrates in deep-sea environments. Through analysis of the genomic data of the hadal snailfish, the authors found that this species may have entered and fully adapted to extreme environments only in the last few million years. Additionally, the study revealed the adaptive features of hadal snailfish in terms of perceptions, circadian rhythms and metabolisms, and the role of ferritin in high-hydrostatic pressure adaptation. Besides, the reads mapping method used to identify events such as gene loss and duplication avoids false positives caused by genome assembly and annotation. This ensures the reliability of the results presented in this manuscript. Overall, these findings provide important clues for a better understanding of deep-sea ecosystems and vertebrate evolution.

Reply: Thank you very much for your positive comments and encouragement.

However, there are some issues that need to be further addressed.1. L119: Please indicate the source of any data used.

Reply: Thank you very much for the suggestion. All data sources used are indicated in Supplementary file 1.

1. L138: The demographic history of hadal snailfish suggests a significant expansion in population size over the last 60,000 years, but the results only show some species, do the results for all individuals support this conclusion?

Reply: Thank you for this suggestion. The estimated demographic history of the hadal snailfish reveals a significant population increase over the past 60,000 years for all individuals. The corresponding results have been incorporated into Figure 1-figure supplements 8B.

**Author response image 1. sa3fig1:** Demographic history for 5 hadal snailfish individuals and 2 Tanaka’s snailfish individuals inferred by PSMC. The generation time of one year for Tanaka snailfish and three years for hadal snailfish.

1. Figure 1-figure supplements 8: Is there a clear source of evidence for the generation time of 1 year chosen for the PSMC analysis?

Reply: We apologize for the inclusion of an incorrect generation time in Figure 1-figure supplements 8. It is important to note that different generation times do not change the shape of the PSMC curve, they only shift the curve along the axis. Due to the absence of definitive evidence regarding the generation time of the hadal snailfish, we have referred to Wang et al., 2019, assuming a generation time of one year for Tanaka snailfish and three years for hadal snailfish. The generation time has been incorporated into the main text (lines 516-517): “The generation time of one year for Tanaka snailfish and three years for hadal snailfish.”.

1. L237: Transcriptomic data suggest that the greatest changes in the brain of hadal snailfish compared to Tanaka's snailfish, what functions these changes are specifically associated with, and how these functions relate to deep-sea adaptation.

Reply: Thank you for this suggestion. Through comparative transcriptome analysis, we identified 3,587 up-regulated genes and 3,433 down-regulated genes in the brains of hadal snailfish compared to Tanaka's snailfish. Subsequently, we conducted Gene Ontology (GO) functional enrichment analysis on the differentially expressed genes, revealing that the up-regulated genes were primarily associated with cilium, DNA repair, protein binding, ATP binding, and microtubule-based movement. Conversely, the down-regulated genes were associated with membranes, GTP-binding, proton transmembrane transport, and synaptic vesicles, as shown in following table (Supplementary file 15). Previous studies have shown that high hydrostatic pressure induces DNA strand breaks and damage, and that DNA repair-related genes upregulated in the brain may help hadal snailfish overcome these challenges.

**Author response table 1. sa3table1:** GO enrichment of expression up-regulated and down-regulated genes in hadal snailfish brain.

	GO term	Type	Function	Adjustedp-value
Up_-_regulated	GO:0005929	Cellular Component	cilium	2.162E-04
	GO:0006281	Biological Process	DNA repair	0.001
	GO:0005515	Molecular Function	protein binding	0.002
	GO:0005524	Molecular Function	ATP binding	0.005
	GO:0007018	Biological Process	microtubule-basedmovement	0.029
Down-regulated	GO:0016020	Cellular Component	membrane	8.799E-13
	GO:0016021	Cellular Component	integral component ofmembrane	2.504E-06
	GO:0005886	Cellular Component	plasma membrane	3.248E-05
	GO:0005525	Molecular Function	GTP binding	1.089E-03
	GO:0003924	Molecular Function	GTPase activity	0.001
	GO: 1902600	Biological Process	proton transmembranetransport	0.003
	GO:0005739	Cellular Component	mitochondrion	0.009
	GO:0005743	Cellular Component	mitochondrial innermembrane	0.010
	GO:0008021	Cellular Component	synaptic vesicle	0.049
	GO:0007399	Biological Process	nervous systemdevelopment	0.049

We have added new results (Supplementary file 15) and descriptions to show the changes in the brains of hadal snailfish (lines 250-255): “Specifically, there are 3,587 up-regulated genes and 3,433 down-regulated genes in the brain of hadal snailfish compared to Tanaka snailfish, and Gene Ontology (GO) functional enrichment analyses revealed that up-regulated genes in the hadal snailfish are associated with cilium, DNA repair, and microtubule-based movement, while down-regulated genes are enriched in membranes, GTP-binding, proton transmembrane transport, and synaptic vesicles (Supplementary file 15).”

1. L276: What is the relationship between low bone mineralization and deep-sea adaptation, and can low mineralization help deep-sea fish better adapt to the deep sea?

Reply: Thank you for this suggestion. The hadal snailfish exhibits lower bone mineralization compared to Tanaka's snailfish, which may have facilitated its adaptation to the deep sea. On one hand, this reduced bone mineralization could have contributed to the hadal snailfish's ability to maintain neutral buoyancy without excessive energy expenditure. On the other hand, the lower bone mineralization may have also rendered their skeleton more flexible and malleable, enhancing their resilience to high hydrostatic pressure. Accordingly, we added the following new descriptions (lines 295-300): “Nonetheless, micro-CT scans have revealed shorter bones and reduced bone density in hadal snailfish, from which it has been inferred that this species has reduced bone mineralization (M. E. Gerringer et al., 2021); this may be a result of lowering density by reducing bone mineralization, allowing to maintain neutral buoyancy without expending too much energy, or it may be a result of making its skeleton more flexible and malleable, which is able to better withstand the effects of HHP.”

1. L293: The abbreviation HHP was mentioned earlier in the article and does not need to be abbreviated here.

Reply: Thank you for the correction. We have corrected the word. Line 315.

1. L345: It should be "In addition, the phylogenetic relationships between different individuals clearly indicate that they have successfully spread to different trenches about 1.0 Mya".

Reply: Thank you for the correction. We have corrected the word. Line 374.

1. It is curious what functions are associated with the up-regulated and down-regulated genes in all tissues of hadal snailfish compared to Tanaka's snailfish, and what functions have hadal snailfish lost in order to adapt to the deep sea?

Reply: Thank you for this suggestion. We added a description of this finding in the results section (lines 337-343): “Next, we identified 34 genes that are significantly more highly expressed in all organs of hadal snailfish in comparison to Tanaka’s snailfish and zebrafish, while only seven genes were found to be significantly more highly expressed in Tanaka’s snailfish using the same criterion (Figure 5-figure supplements 1). The 34 genes are enriched in only one GO category, GO:0000077: DNA damage checkpoint (Adjusted P-value: 0.0177). Moreover, five of the 34 genes are associated with DNA repair.” This suggests that up-regulated genes in all tissues in hadal snailfish are associated with DNA repair in response to DNA damage caused by high hydrostatic pressure, whereas down-regulated genes do not show enrichment for a particular function.

Overall, the functions lost in hadal snailfish adapted to the deep sea are mainly related to the effects of the dark environment, which can be summarized as follows (lines 375-383): “The comparative genomic analysis revealed that the complete absence of light had a profound effect on the hadal snailfish. In addition to the substantial loss of visual genes and loss of pigmentation, many rhythm-related genes were also absent, although some rhythm genes were still present. The gene loss may not only come from relaxation of natural selection, but also for better adaptation. For example, the grpr gene copies are absent or down-regulated in hadal snailfish, which could in turn increased their activity in the dark, allowing them to survive better in the dark environment (Wada et al., 1997). The loss of gpr27 may also increase the ability of lipid metabolism, which is essential for coping with short-term food deficiencies (Nath et al., 2020).”

**Reviewer #2 (Recommendations For The Authors):**
I have pointed out some of the examples that struck me as worthy of additional thought/writing/comments from the authors. Any changes/comments are relatively minor.

Reply: Thank you very much for your positive comments on this work.

For comparative transcriptome analyses, reads were mapped back to reference genomes and TPM values were obtained for gene-level count analyses. 1:1 orthologs were used for differential expression analyses. This is indeed the only way to normalize counts across species, by comparing the same gene set in each species. Differential expression statistics were run in DEseq2. This is a robust way to compare gene expression across species and where fold-change values are reported (e.g. Fig 3, creatively by coloring the gene name) the values are best-practice.In other places, TPM values are reported (e.g. Fig 2D, Fig 4C, Fig 5A, Fig 4-Fig supp 4) to illustrate expression differences within a tissue across species. The comparisons look robust, although it is not made clear how the values were obtained in all cases. For example, in Fig 2D the TPM values appear to be from eyes of individual fish, but in Fig 4C and 5A they must be some kind of average? I think that information should be added to the figure legends.Of note: TPM values are sensitive to the shape of the RNA abundance distribution from a given sample: A small number of very highly expressed genes might bias TPM values downward for other genes. From one individual to another or from one species to another, it is not obvious to me that we should expect the same TPM distribution from the same tissues, making it a challenging metric for comparison across samples, and especially across species. An alternative measure of RNA abundance is normalized counts that can be output from DEseq2. See:Zhao, Y., Li, M.C., Konaté, M.M., Chen, L., Das, B., Karlovich, C., Williams, P.M., Evrard, Y.A., Doroshow, J.H. and McShane, L.M., 2021. TPM, FPKM, or normalized counts? A comparative study of quantification measures for the analysis of RNA-seq data from the NCI patient-derived models repository. Journal of translational medicine, 19(1), pp.1-15.If the authors would like to keep the TPM values, I think it would be useful for them to visualize the TPM value distribution that the numbers were derived from. One way to do this would be to make a violin plot for species/tissue and plot the TPM values of interest on that. That would give a visualization of the ranked value of the gene within the context of all other TPM values. A more highly expressed gene would presumably have a higher rank in context of the specific tissue/species and be more towards the upper tail of the distribution. An example violin plot can be found in Fig 6 of:Burns, J.A., Gruber, D.F., Gaffney, J.P., Sparks, J.S. and Brugler, M.R., 2022. Transcriptomics of a Greenlandic Snailfish Reveals Exceptionally High Expression of Antifreeze Protein Transcripts. Evolutionary Bioinformatics, 18, p.11769343221118347.Alternatively, a comparison of TPM and normalized count data (heatmaps?) would be of use for at least some of the reported TPM values to show whether the different normalization methods give comparable outputs in terms of differential expression. One reason for these questions is that DEseq2 uses normalized counts for statistical analyses, but values are expressed as TPM in the noted figures (yes, TPM accounts for transcript length, but can still be subject to distribution biases).

Reply: Thank you for your suggestions. Following your suggestions, we modified Fig 2D, Fig 4C, Fig 4-Fig supp 4, and Fig 5-Fig supp 1, respectively. In the differential expression analyses, only one-to-one orthologues of hadal snailfish and Tanaka's snailfish can get the normalized counts output by DEseq2, so we showed the normalized counts by DEseq2 output for Fig 2D, Fig 4C, Fig 4-Fig supp 4, Fig 5-Fig supp 1, and for Fig 5A, since the copy number of fthl27 genes undergoes specific expansion in hadal snailfish, we visualized the ranking of all fthl27 genes across tissues by plotting violins in Fig 5-Fig supp 2.

**Author response image 2. sa3fig2:** Log10-transformation normalized counts for DESeq2 (COUNTDESEQ2) of vision-related genes in the eyes of hadal snailfish and Tanka's snailfish. * represents genes significantly downregulated in hadal snailfish (corrected P < 0.05).

**Author response image 3. sa3fig3:** The deletion of one copy of grpr and another copy of down-regulated expression in hadal snailfish. The relative positions of genes on chromosomes are indicated by arrows, with arrows to the right representing the forward strand and arrows to the left representing the reverse strand. The heatmap presented is the average of the normalized counts for DESeq2 (COUNTDESEQ2) in all replicate samples from each tissue. * represents tissue in which the grpr-1 was significantly down-regulated in hadal snailfish (corrected P < 0.05).

**Author response image 4. sa3fig4:** Expression of the vitamin D related genes in various tissues of hadal snailfish and Tanaka's snailfish. The heatmap presented is the average of the normalized counts for DESeq2 (COUNTDESEQ2) in all replicate samples from each tissue.

**Author response image 5. sa3fig5:** Expression of the ROS-related genes in different tissues of hadal snailfish and Tanaka's snailfish. The heatmap presented is the average of the normalized counts for DESeq2 (COUNTDESEQ2) in all replicate samples from each tissue.

**Author response image 6. sa3fig6:** Ranking of the expression of individual copies of fthl27 gene in hadal snailfish and Tanaka's snailfish in various tissues showed that all copies of fthl27 in hadal snailfish have high expression. The gene expression presented is the average of TPM in all replicate samples from each tissue.

Line 96: Which BUSCOs? In the methods it is noted that the actinopterygii_odb10 BUSCO set was used. I think it should also be noted here so that it is clear which BUSCO set was used for completeness analysis. It could even be informally the ray-finned fish BUSCOs or Actinopterygii BUSCOs.

Reply: Thank you for this suggestion. We used Actinopterygii_odb10 database and we added the BUSCO set to the main text as follows (lines 92-95): “The new assembly filled 1.26 Mb of gaps that were present in our previous assembly and have a much higher level of genome continuity and completeness (with complete BUSCOs of 96.0 % [Actinopterygii_odb10 database]) than the two previous assemblies.”

Lines 102-105: The medaka genome paper proposes the notion that the ancestral chromosome number between medaka, tetraodon, and zebrafish is 24. There may be other evidence of that too. Some of that evidence should be cited here to support the notion that sticklebacks had chromosome fusions to get to 21 chromosomes rather than scorpionfish having chromosome fissions to get to 24. Here's the medaka genome paper:Kasahara, M., Naruse, K., Sasaki, S., Nakatani, Y., Qu, W., Ahsan, B., Yamada, T., Nagayasu, Y., Doi, K., Kasai, Y. and Jindo, T., 2007. The medaka draft genome and insights into vertebrate genome evolution. Nature, 447(7145), pp.714-719.

Reply: Thank you for your great suggestion. Accordingly, we modified the sentence and added the citation as follows (lines 100-105): “We noticed that there is no major chromosomal rearrangement between hadal snailfish and Tanaka’s snailfish, and chromosome numbers are consistent with the previously reported MTZ-ancestor (the last common ancestor of medaka, Tetraodon, and zebrafish) (Kasahara et al., 2007), while the stickleback had undergone several independent chromosomal fusion events (Figure 1-figure supplements 4).”

Line 161-173: "Along with the expression data, we noticed that these genes exhibit a different level of relaxation of natural selection in hadal snailfish (Figure 2B; Figure 2-figure supplements 1)." With the above statment and evidence, the authors are presumably referring to gene losses and differences in expression levels. I think that since gene expression was not measured in a controlled way it may not be a good measure of selection throughout. The reported genes could be highly expressed under some other condition, selection intact. I find Fig2-Fig supp 1 difficult to interpret. I assume I am looking for regions where Tanaka’s snailfish reads map and Hadal snailfish reads do not, but it is not abundantly clear. Also, other measures of selection might be good to investigate: accumulation of mutations in the region could be evidence of relaxed selection, for example, where essential genes will accumulate fewer mutations than conditional genes or (presumably) genes that are not needed at all. The authors could complete a mutational/SNP analysis using their genome data on the discussed genes if they want to strengthen their case for relaxed selection. Here is a reference (from Arabidopsis) showing these kinds of effects:Monroe, J.G., Srikant, T., Carbonell-Bejerano, P., Becker, C., Lensink, M., Exposito-Alonso, M., Klein, M., Hildebrandt, J., Neumann, M., Kliebenstein, D. and Weng, M.L., 2022. Mutation bias reflects natural selection in *Arabidopsis thaliana*. Nature, 602(7895), pp.101-105.

Reply: Thank you for pointing out this important issue. Following your suggestion, we have removed the mention of the down-regulation of some visual genes in the eyes of hadal snailfish and the results of the original Fig2-Fig supp 1 that were based on reads mapping to confirm whether the genes were lost or not. To investigate the potential relaxation of natural selection in the opn1sw2 gene in hadal snailfish, we conducted precise gene structure annotation. Our findings revealed that the opn1sw2 gene is pseudogenized in hadal snailfish, indicating a relaxation of natural selection. We have included this result in Figure 2-figure supplements 1.

**Author response image 7. sa3fig7:** Pseudogenization of opn1sw2 in hadal snailfish. The deletion changed the protein’s sequence, causing its premature termination.

Accordingly, we have toned down the related conclusions in the main text as follows (lines 164-173): “We noticed that the lws gene (long wavelength) has been completely lost in both hadal snailfish and Tanaka’s snailfish; rh2 (central wavelength) has been specifically lost in hadal snailfish (Figure 2B and 2C); sws2 (short wavelength) has undergone pseudogenization in hadal snailfish (Figure 2-figure supplements 1); while rh1 and gnat1 (perception of very dim light) is both still present and expressed in the eyes of hadal snailfish (Figure 2D). A previous study has also proven the existence of rhodopsin protein in the eyes of hadal snailfish using proteome data (Yan, Lian, Lan, Qian, & He, 2021). The preservation and expression of genes for the perception of very dim light suggests that they are still subject to natural selection, at least in the recent past.”

Line 161-170: What tissue were the transcripts derived from for looking at expression level of opsins? Eyes?

Reply: Thank you for your suggestions. The transcripts used to observe the expression levels of optic proteins were obtained from the eye.

Line 191: What does tmc1 do specifically?

Reply: Thank you for this suggestion. The tmc1 gene encodes transmembrane channel-like protein 1, involved in the mechanotransduction process in sensory hair cells of the inner ear that facilitates the conversion of mechanical stimuli into electrical signals used for hearing and homeostasis. We added functional annotations for the tmc1 in the main text (lines 190-196): “Of these, the most significant upregulated gene is tmc1, which encodes transmembrane channel-like protein 1, involved in the mechanotransduction process in sensory hair cells of the inner ear that facilitates the conversion of mechanical stimuli into electrical signals used for hearing and homeostasis (Maeda et al., 2014), and some mutations in this gene have been found to be associated with hearing loss (Kitajiri, Makishima, Friedman, & Griffith, 2007; Riahi et al., 2014).”

Line 208: "it is likely" is a bit proscriptive

Reply: Thank you for this suggestion. We rephrased the sentence as follows (lines 213-215): “Expansion of cldnj was observed in all resequenced individuals of the hadal snailfish (Supplementary file 10), which provides an explanation for the hadal snailfish breaks the depth limitation on calcium carbonate deposition and becomes one of the few species of teleost in hadal zone.”

Line 199: maybe give a little more info on exactly what cldnj does? e.g. "cldnj encodes a claudin protein that has a role in tight junctions through calcium independent cell-adhesion activity" or something like that.

Reply: Thank you for this suggestion. We have added functional annotations for the cldnj to the main text (lines 200-204): “Moreover, the gene involved in lifelong otolith mineralization, cldnj, has three copies in hadal snailfish, but only one copy in other teleost species, encodes a claudin protein that has a role in tight junctions through calcium independent cell-adhesion activity (Figure 3B, Figure 3C) (Hardison, Lichten, Banerjee-Basu, Becker, & Burgess, 2005).”

Lines 199-210: Paragraph on cldnj: there are extra cldnj genes in the hadal snailfish, but no apparent extra expression. Could the authors mention that in their analysis/discussion of the data?

Reply: Thank you for your suggestions. Despite not observing significant changes in cldnj expression in the brain tissue of hadal snailfish compared to Tanaka's snailfish, it is important to consider that the brain may not be the primary site of cldnj expression. Previous studies in zebrafish have consistently shown expression of cldnj in the otocyst during the critical early growth phase of the otolith, with a lower level of expression observed in the zebrafish brain. However, due to the unavailability of otocyst samples from hadal snailfish in our current study, our findings do not provide confirmation of any additional expression changes resulting from cldnj amplification. Consequently, it is crucial to conduct future comprehensive investigations to explore the expression patterns of cldnj specifically in the otocyst of hadal snailfish. Accordingly, we added a discussion of this result in the main text (lines 209-214): “In our investigation, we found that the expression of cldnj was not significantly up-regulated in the brain of the hadal snailfish than in Tanaka’s snailfish, which may be related to the fact that cldnj is mainly expressed in the otocyst, while the expression in the brain is lower. However, due to the immense challenge in obtaining samples of hadal snailfish, the expression of cldnj in the otocyst deserves more in-depth study in the future.”

Lines 225-231: I wonder whether low expression of a circadian gene might be a time of day effect rather than an evolutionary trait. Could the authors comment?

Reply: Thank you for your suggestions. Previous studies have shown that the grpr gene is expressed relatively consistently in mouse suprachiasmatic nucleus (SCN) throughout the day (Figure 4-figure supplements 1) and we hypothesize that the low expression of grpr-1 gene expression in hadal snailfish is an evolutionary trait. We have modified this result in the main text (lines 232-242): “In addition, in the teleosts closely related to hadal snailfish, there are usually two copies of grpr encoding the gastrin-releasing peptide receptor; we noticed that in hadal snailfish one of them is absent and the other is barely expressed in brain (Figure 4C), whereas a previous study found that the grpr gene in the mouse suprachiasmatic nucleus (SCN) did not fluctuate significantly during a 24-hour light/dark cycle and had a relatively stable expression (Pembroke, Babbs, Davies, Ponting, & Oliver, 2015) (Figure 4-figure supplements 1). It has been reported that grpr deficient mice, while exhibiting normal circadian rhythms, show significantly increased locomotor activity in dark conditions (Wada et al., 1997; Zhao et al., 2023). We might therefore speculate that the absence of that gene might in some way benefit the activity of hadal snailfish under complete darkness.”

**Author response image 8. sa3fig8:** Expression of the grpr in a 24-hour light/dark cycle in the mouse suprachiasmatic nucleus (SCN). Data source with http://www.wgpembroke.com/shiny/SCNseq.

Line 253: What is gpr27? G protein coupled receptor?

Reply: We apologize for the ambiguous description. Gpr27 is a G protein-coupled receptor, belonging to the family of cell surface receptors. We introduced gpr27 in the main text as follows (lines 270-273): “Gpr27 is a G protein-coupled receptor, belonging to the family of cell surface receptors, involved in various physiological processes and expressed in multiple tissues including the brain, heart, kidney, and immune system.”

Line 253: Fig4 Fig supp 3 is a good example of pseudogenization!

Reply: Thank you very much for your recognition.

Line 279: What is bglap? It regulates bone mineralization, but what specifically does that gene do?

Reply: We apologize for the ambiguous description. The bglap gene encodes a highly abundant bone protein secreted by osteoblasts that binds calcium and hydroxyapatite and regulates bone remodeling and energy metabolism. We introduced bglap in the main text as follows (lines 300-304): “The gene bglap, which encodes a highly abundant bone protein secreted by osteoblasts that binds calcium and hydroxyapatite and regulates bone remodeling and energy metabolism, had been found to be a pseudogene in hadal fish (K. Wang et al., 2019), which may contribute to this phenotype.”

Line 299: Introduction of another gene without providing an exact function: acaa1.

Reply: We apologize for the ambiguous description. The acaa1 gene encodes acetyl-CoA acetyltransferase 1, a key regulator of fatty acid β-oxidation in the peroxisome, which plays a controlling role in fatty acid elongation and degradation. We introduced acaa1 in the main text as follows (lines 319-324): “In regard to the effect of cell membrane fluidity, relevant genetic alterations had been identified in previous studies, i.e., the amplification of acaa1 (encoding acetyl-CoA acetyltransferase 1, a key regulator of fatty acid β-oxidation in the peroxisome, which plays a controlling role in fatty acid elongation and degradation) may increase the ability to synthesize unsaturated fatty acids (Fang et al., 2000; K. Wang et al., 2019).”

Fig 5 legend: The DCFH-DA experiment is not an immunofluorescence assay. It is better described as a redox-sensitive fluorescent probe. Please take note throughout.

Reply: Thank you for pointing out our mistakes. We corrected the word. Line 1048 and 1151 as follows: “ROS levels were confirmed by redox-sensitive fluorescent probe using DCFH-DA molecular probe in 293T cell culture medium with or without fthl27-overexpression plasmid added with H2O2 or FAC for 4 hours.”

Line 326: Manuscript notes that ROS levels in transfected cells are "significantly lower" than the control group, but there is no quantification or statistical analysis of ROS levels. In the methods, I noticed the mention of flow cytometry, but do not see any data from that experiment. Proportion of cells with DCFH-DA fluorescence above a threshold would be a good statistic for the experiment... Another could be average fluorescence per cell. Figure 5B shows some images with green dots and it looks like more green in the "control" (which could better be labeled as "mock-transfection") than in the fthl27 overexpression, but this could certainly be quantified by flow cytometry. I recommend that data be added.

Reply: Thank you for your suggestions. We apologize for the error in the main text, we used a fluorescence microscope to observe fluorescence in our experiments, not a flow cytometer. We have corrected it in the methods section as follows (lines 651-653): “ROS levels were measured using a DCFH-DA molecular probe, and fluorescence was observed through a fluorescence microscope with an optional FITC filter, with the background removed to observe changes in fluorescence.” Meanwhile, we processed the images with ImageJ to obtain the respective mean fluorescence intensities (MFI) and found that the MFI of the fthl27-overexpression cells were lower than the control group, which indicated that the ROS levels of the fthl27-overexpression cells were significantly lower than the control group. MFI has been added to Figure 5B.

**Author response image 9. sa3fig9:** ROS levels were confirmed by redox-sensitive fluorescent probe using DCFH-DA molecular probe in 293T cell culture medium with or without fthl27-overexpression plasmid added with H2O2 or FAC for 4 hours. Images are merged from bright field images with fluorescent images using ImageJ, while the mean fluorescence intensity (MFI) is also calculated using ImageJ. Green, cellular ROS. Scale bars equal 100 μm.

Regarding the ROS experiment: Transfection of HEK293T cells should be reasonably straightforward, and the experiment was controlled appropriately with a mock transfection, but some additional parameters are still needed to help interpret the results. Those include: Direct evidence that the transfection worked, like qPCR, western blots (is the fthl27 tagged with an antigen?), coexpression of a fluorescent protein. Then transfection efficiency should be calculated and reported.

Reply: Thank you for your suggestions. To assess the success of the transfection, we randomly selected a subset of fthl27-transfected HEK293T cells for transcriptome sequencing. This approach allowed us to examine the gene expression profiles and confirm the efficacy of the transfection process. As control samples, we obtained transcriptome data from two untreated HEK293T cells (SRR24835259 and SRR24835265) from NCBI. Subsequently, we extracted the fthl27 gene sequence of the hadal snailfish, along with 1,000 bp upstream and downstream regions, as a separate scaffold. This scaffold was then merged with the human genome to assess the expression levels of each gene in the three transcriptome datasets. The results demonstrated that the fthl27 gene exhibited the highest expression in fthl27-transfected HEK293T cells, while in the control group, the expression of the fthl27 gene was negligible (TPM = 0). Additionally, the expression patterns of other highly expressed genes were similar to those observed in the control group, confirming the successful fthl27 transfection. These findings have been incorporated into Figure 5-figure supplements 3.

**Author response image 10. sa3fig10:** (B) Reads depth of fthl27 gene in fthl27-transfected HEK293T cells and 2 untreated HEK293T cells (SRR24835259 and SRR24835265) transcriptome data. (C) Expression of each gene in the transcriptome data of fthl27-transfected HEK293T cells and 2 untreated HEK293T cells (SRR24835259 and SRR24835265), where the genes shown are the 4 most highly expressed genes in each sample.

Lines 383-386: expression of DNA repair genes is mentioned, but not shown anywhere in the results?

Reply: Thank you for your suggestions. Accordingly, we added a description of this finding in the results section (lines 337-343): “Next, we identified 34 genes that are significantly more highly expressed in all organs of hadal snailfish in comparison to Tanaka’s snailfish and zebrafish, while only seven genes were found to be significantly more highly expressed in Tanaka’s snailfish using the same criterion (Figure 5-figure supplements 1). The 34 genes are enriched in only one GO category, GO:0000077: DNA damage checkpoint (Adjusted P-value: 0.0177). Moreover, five of the 34 genes are associated with DNA repair.”. And we added the information in the Figure 5-figure supplements 1C.

**Author response image 11. sa3fig11:** Genes were significantly more highly expressed in all tissues of the hadal snailfish compared to Tanaka's snailfish, and 5 genes (purple) were associated with DNA repair.